# Neural dynamics of reversal learning in the prefrontal cortex and recurrent neural networks

**Christopher M Kim[1]\*[†], Carson C Chow[1], Bruno B Averbeck[1,2]**

[1]Laboratory of Biological Modeling, National Institute of Diabetes and Digestive and Kidney Diseases, National Institutes of Health, Bethesda, United States; [2]Laboratory of Neuropsychology, National Institute of Mental Health, National Institutes of Health, Bethesda, United States

**\*For correspondence:** chrismkkim@gmail.com

**Present address:** [†]Department of Mathematics, Howard University, Washington, DC, United States

**Competing interest:** The authors declare that no competing interests exist.

**Preprint posted** 15 September 2024

**Sent for Review** 04 October 2024

**Reviewed preprint posted** 06 December 2024

**Reviewed preprint revised** 10 June 2025

**Version of Record published** 23 September 2025

## eLife Assessment

The findings of this study are **valuable**, offering insights into the neural representation of reversal probability in decision-making tasks, with potential implications for understanding flexible behavior in changing environments. The study contains interesting comparisons between neural data and models, including evidence for partial consistency with line attractor models in this probabilistic reversal learning task. However, it remains **incomplete** due to issues related to how the RNN training and the analysis of its dynamics, which renders the evidence as not complete.

**Abstract** In probabilistic reversal learning, the choice option yielding reward with higher probability switches at a random trial. To perform optimally in this task, one has to accumulate evidence across trials to infer the probability that a reversal has occurred. We investigated how this reversal probability is represented in cortical neurons by analyzing the neural activity in the prefrontal cortex of monkeys and recurrent neural networks trained on the task. We found that in a neural subspace encoding reversal probability, its activity represented integration of reward outcomes as in a line attractor model. The reversal probability activity at the start of a trial was stationary, stable, and consistent with the attractor dynamics. However, during the trial, the activity was associated with task-related behavior and became non-stationary, thus deviating from the line attractor. Fitting a predictive model to neural data showed that the stationary state at the trial start serves as an initial condition for launching the non-stationary activity. This suggested an extension of the line attractor model with behavior-induced non-stationary dynamics. The non-stationary trajectories were separable indicating that they can represent distinct probabilistic values. Perturbing the reversal probability activity in the recurrent neural networks biased choice outcomes demonstrating its functional significance. In sum, our results show that cortical networks encode reversal probability in stable stationary state at the start of a trial and utilize it to initiate non-stationary dynamics that accommodates task-related behavior while maintaining the reversal information.

## Introduction

To survive in a dynamically changing world, animals must interact with the environment and learn from their experience to adjust their behavior. Reversal learning has been used for assessing the ability to adapt one's behavior in such an environment (*Butter, 1969*; *Costa et al., 2015*; *Groman et al., 2019*; *Bartolo and Averbeck, 2020*; *Su and Cohen, 2022*; *Hyun et al., 2023*). For instance, in two-armed bandit tasks with probabilistic reward, the subject learns from initial trials that one option has higher

reward probability than the other. When the reward probability of the two options is reversed at a random trial, the subject must learn to reverse its preferred choice to maximize reward outcome. In these tasks, there is uncertainty in when to reverse one's choice, as reward is received stochastically even when the less favorable option is chosen. Therefore, it is essential that reward outcomes are integrated over multiple trials before the initial choice preference is reversed. Although neural mechanisms for accumulating evidence within a trial have been studied extensively (*Wang, 2002*; *Inagaki et al., 2019*; *Luo et al., 2023*; *Sutton, 1988*), it remains unclear if a recurrent neural circuit uses a similar neural mechanism for accumulating evidence across multiple trials, while performing task-related intervening behavior during each trial.

In this study, we merged two classes of computational models, i.e., behavioral and neural, to investigate the neural basis of multi-trial evidence accumulation. Behavior models capture subject's behavioral strategies for performing the reversal learning task. For instance, model-free reinforcement learning (RL) (*Rescorla, 1972*; *Averbeck, 2017*; *Jang et al., 2015*) assumes that the subject learns only from choices and reward outcomes without specific knowledge about task structure. Model-based Bayesian inference (*Costa et al., 2015*; *Wilson et al., 2010*; *Wong and Wang, 2006*), in contrast, assumes that the task structure is known to the subject, and one can infer reversal points statistically, resulting in abrupt switches in choice preference. Model-based and model-free RL models are formal models that do not specify implementation in a network of neurons. On the other hand, neural models implemented with recurrent neural networks (RNNs) can be trained to use recurrent activity to perform the reversal learning task. In particular, attractor dynamics, in which the network state moves towards discrete (*Luo et al., 2023*; *Seung, 1996*) or along continuous (*Inagaki et al., 2019*; *Bollimunta et al., 2012*) attractor states, have been studied extensively as a potential neural mechanism for decision-making and evidence accumulation (*Brody and Hanks, 2016*; *Wang et al., 2018*).

We sought to train continuous time RNNs to mimic the behavioral strategies of monkeys performing the reversal learning task. Previous studies (*Costa et al., 2015*; *Bartolo and Averbeck, 2020*) have shown that a Bayesian inference model can capture a key aspect of the monkey's behavioral strategy, i.e., adhere to the preferred choice until the reversal of reward schedule is detected and then switch abruptly. We trained the RNNs to replicate this behavioral strategy by training them on target behaviors generated from the Bayesian model.

We found that the activity in the neural subspace representing reversal probability could be explained by integrating reward outcomes across trials. At the start of a trial, when the subject was holding fixation before cues were shown, the reversal probability activity was stationary and stable. This stationary activity mode was compatible with the line attractor model that accumulates evidence along attracting states. However, during the trial, the neural activity representing reversal probability had substantial dynamics and was associated with task-related behavior, such as making decisions or receiving feedback. The non-stationary activity during the trial made the line attractor model, which requires the network state to stay close to attractor states (*Inagaki et al., 2019*; *Luo et al., 2023*; *Seung, 1996*; *Bollimunta et al., 2012*), inadequate for explaining the neural activity encoding evidence accumulation in reversal learning.

To better understand how reversal probability is represented in two different activity modes, we investigated the underlying dynamics that link the two. We found that the non-stationary trajectory can be predicted from the stationary activity at the trial start, suggesting that underlying dynamics, associated with task-related behavior, generates the non-stationary activity. In addition, separable points at the initial state remained separable as they propagated in time, allowing distinct probabilistic values to be represented in the trajectories. These findings suggested an extension of the line attractor model where stationary activity on the line attractor provides an initial state from which non-stationary dynamics that preserves separability is initiated. Finally, perturbing reversal probability activity causally affected choice outcomes, demonstrating its functional significance.

Our results show that, in a probabilistic reversal learning task, cortical networks encode the reversal probability by adopting, not only stationary states as in a line attractor, but also separable dynamic trajectories that can represent distinct probabilistic values and accommodate non-stationary dynamics associated with task-related behavior.

# Results

## Trained RNN's choices are consistent with monkey behavior

We considered a reversal learning task that monkeys were trained to perform in a previous study (*Bartolo and Averbeck, 2020*). In our study, we trained the RNNs to perform a task similar to the task the monkeys performed. A reversal learning task was completed by executing a block of a fixed number of trials. In each trial, the subject (monkey or RNN) chose one of the two available options, and reward for the choice was delivered stochastically. At the beginning of a block, one option was designated as the high-value option, while the other option was designated as the low-value option. The high-value option was rewarded 70% of the time when chosen, and the low-value option was rewarded 30% of the time when chosen. The reward probability of two options was switched at a random trial near the mid-trial of a block, and the reversed reward schedule was maintained until the end of the block. The reversal of reward schedule occurred only once in our task, differing from other versions of reversal learning where reversal occurs multiple times across trials (*Su and Cohen, 2022*; *Wang et al., 2018*). The subject had to learn to switch its preferred choice to maximize reward. Because reward delivery was stochastic, the subject had to infer the reversal of reward schedule by accumulating evidence that a reversal had occurred.

The sequence of events occurring in a trial was structured similarly in the RNN and the monkey tasks. As described in *Bartolo and Averbeck, 2020*, in each trial, the monkeys were first shown a signal that required them to fixate for a variable time (400 - 800 ms). Then, two cues were presented to both sides of the fixation dot simultaneously. The monkeys made a saccade to an option to report their choice. After holding for 500 ms, the reward was delivered (see Methods Experiment setup for monkeys for details). In the RNNs, they were first stimulated by a cue signaling them to make a choice. After a short delay, the RNNs made a choice between two options. Then, the RNNs received feedback that signaled the choice it made and whether the outcome was rewarded (*Figure 1A*). This same trial structure was repeated across all trials in a block (see Methods Recurrent neural network for details).

The RNN and the monkeys tasks had differences. For the RNNs, there were 36 trials in a block during testing (24 trials were used for training) and, for the monkeys, there were 80 trials in a block. The number of RNN trials was reduced to avoid GPU memory overflow issues when training with backpropagation-through-time. For the RNNs, the reversal of reward schedule occurred on a trial randomly chosen from the 10 middle trials of a block. For the monkeys, the reversal trial was chosen randomly from 20 middle trials of a block. The fixation period was not required in the RNNs since they performed the task without it; however, we examined the effects of having a fixation period on neural dynamics (see Figure 4D). In total, 40 successfully trained RNNs were analyzed in our study, where each RNN performed 20 blocks of reversal learning tasks. On the other hand, two monkeys performed in total of eight sessions of experiments, where each session consisted of 24 blocks of reversal learning.

To train RNNs, we set up a reward schedule where the high-value option at the start of a trial was randomly selected from two options. Since the initial high-value option was switched to the other option at a random trial (*Figure 1*, scheduled reversal), the RNNs had to learn to reverse their preferred choice to maximize reward. To learn to reverse, the RNNs were trained to mimic the outputs of a Bayesian inference model that was shown to capture the monkey's reversal behavior in previous studies (*Costa et al., 2015*; *Bartolo and Averbeck, 2020*). We first let the RNNs perform the task by simulating the RNN dynamics starting at a random initial state and providing relevant stimuli, such as cue and feedback, at every trial. Once the RNNs completed a block of trials, the choice and reward outcomes of all trials in a block were fed into the Bayesian model to infer the trial at which reward schedule was reversed, referred to as the inferred scheduled reversal trial (*Figure 1B*, inferred reversal). Previous studies have shown that monkeys reverse their preferred choice (*Figure 1B*, behavioral reversal) a few trials after the scheduled reversal trial (*Costa et al., 2015*; *Bartolo and Averbeck, 2020*). Therefore, the target choice outputs (*Figure 1B*, target output), on which the RNNs were trained, were set to be the initial high-value option until a few trials after the inferred scheduled reversal trial, followed by an abrupt switch to the other option. The recurrent weights and the output weights of the RNNs were trained via supervised learning to minimize the cross-entropy between the RNN choice outputs and the target choice outputs (See Methods Training scheme for details on the RNN training).

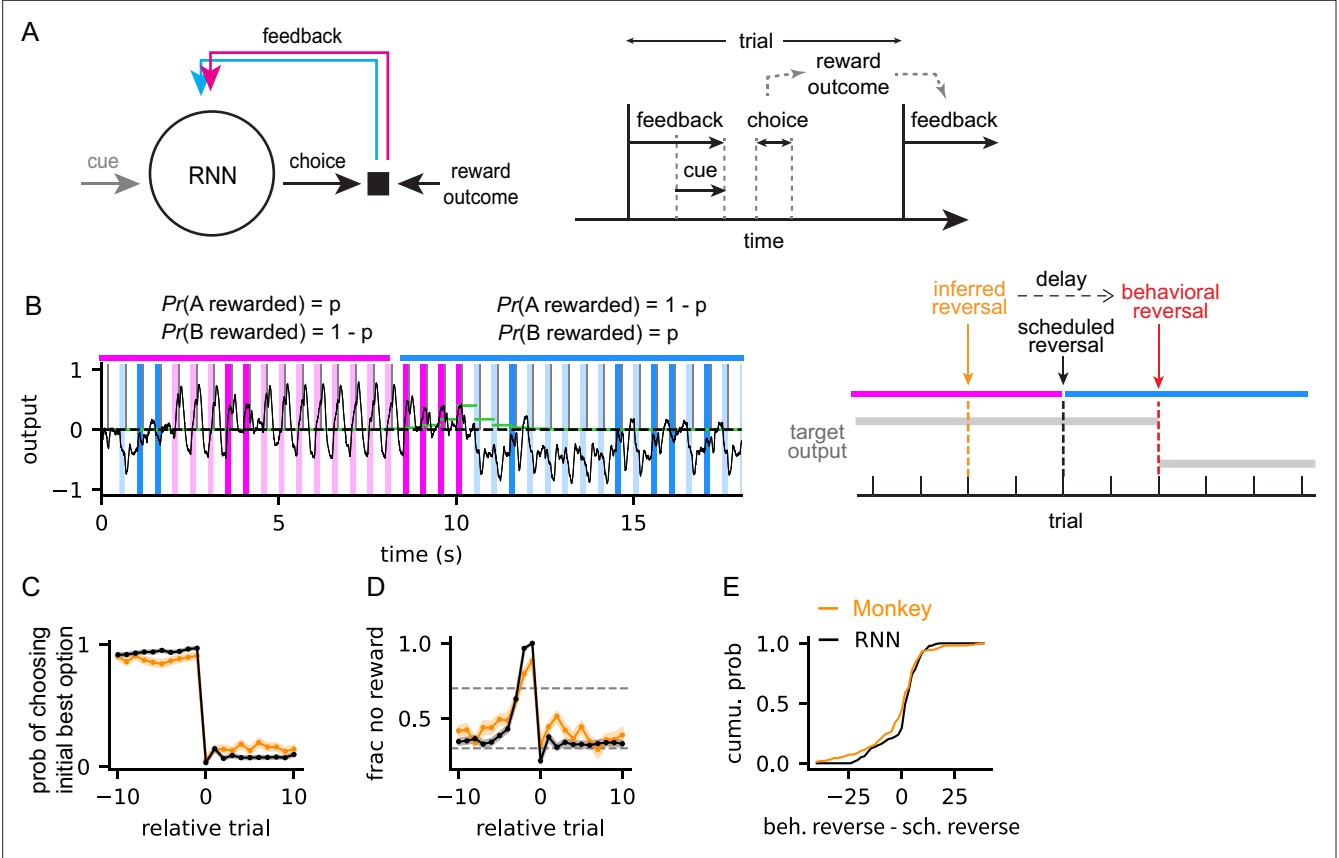

**Figure 1.** Comparison of the behavior of trained recurrent neural networks (RNNs) and monkeys. (**A**) Schematic of RNN training setup. In a trial, the network makes a choice in response to a cue. Then, a feedback input, determined by the choice and reward outcome, is injected to the network. This procedure is repeated across trials. The panel on the right shows this sequence of events unfolding in time in a trial. (**B**) Left: Example of a trained RNN's choice outcomes. Vertical bars show RNN choices in each trial and the reward outcomes (magenta: choice A, blue: choice B, light: rewarded, dark: not rewarded). Horizontal bars on the top show reward schedules (magenta: choice A receiving reward is 70%, choice B receiving reward is 30%; blue: reward schedule is reversed). Black curve shows the RNN output. Green horizontal bars show the posterior of reversal probability at each trial inferred using Bayesian model. Right: Schematic of RNN training scheme. The scheduled reversal indicates the trial at which the reward probabilities of two options switch (color codes for magenta and cyan are the same as the left panel). The inferred reversal is the scheduled reversal trial inferred from the Bayesian model. The behavioral reversal is determined by adding a few delay trials to the inferred reversal trial. The target output, on which the RNNs outputs are trained, switches at the behavioral reversal trial. (**C**) Probability of choosing the initial best (i.e. high-value) option. Relative trial indicates the trial number relative to the behavioral reversal trial inferred from the Bayesian model. Relative trial number 0 is the trial at which the choice was reversed. Shaded region shows the S.E.M (standard error of mean) over blocks in all the sessions (monkeys) or networks (RNNs). (**D**) Fraction of no-reward blocks as a function of relative trial. Dotted lines show 0.3 and 0.7. Shaded region shows the S.E.M (standard error of mean) over blocks in all the sessions (monkeys) or networks (RNNs). (**E**) Distribution of RNN's and monkey's reversal trial, relative to the experimentally scheduled reversal trial.

After training, in a typical block, a trained RNN selected the initial high-value option, despite occasionally not receiving a reward, but abruptly switched its choice when consecutive no-reward trials persisted (*Figure 1B*, left). Such abrupt reversal behavior was expected as the RNNs were trained to mimic the target outputs of the Bayesian inference model (*Figure 1B*, right). The intrinsic time scale of the RNN ($\tau$ =20 ms in *Equation 1* in Methods Recurrent neural network) was substantially faster than the duration of a trial (500 ms), thus the persistent behavior over multiple trials was a result of learning the task. Analyzing the choice outputs of all the trained RNNs showed that, as in the example discussed, they selected the high-value option with high probability before the behavioral reversal, at which time they abruptly switched their choice (*Figure 1C*, black). This abrupt reversal behavior was also found in the monkey's behavior trained on the same task (*Figure 1C*, orange). The behavioral reversal was preceded by a gradually increasing number of no-reward trials in the RNNs and the monkeys (*Figure 1D*). The distribution of behavioral reversal trials (i.e. trial at which preferred choice

was reversed) relative to the scheduled reversal trial (i.e. trial at which reward schedule was reversed) was in good agreement with the distribution of monkey's reversal trials (*Figure 1E*).

## Task-relevant neural activity evolves dynamically

Next, we analyzed the neural activity of neurons in the dorsolateral prefrontal cortex (PFC) of two monkeys and the activation of population of neurons in the RNNs. The spiking activity of PFC neurons was recorded while they performed the reversal learning task using eight microelectrode arrays. In each session, we recorded simultaneously from 573 to 1023 neurons (n=8 sessions, median population size 706). This neural data was collected and analyzed in a previous manuscript (*Bartolo and Averbeck, 2020*). For each PFC neuron, we counted the spikes it emitted in a 300 ms time window that moved in 20 ms increment to analyze its spiking activity over time. For the RNNs, 40 successfully trained RNNs, whose reward rates were close to 70%, were included in the analysis.

We examined the temporal dynamics of task-relevant neural activity, in particular activity encoding the choice and reversal probability. To capture task-relevant neural activity, we first identified population vectors that encoded the task variables using a method called targeted dimensionality reduction (*Mante et al., 2013*). It regresses the activity of each neuron at each time bin onto task variables across trials and finds the maximal population vector of each task variable. Then, neural activity representing the task variable is obtained by projecting the population activity onto the identified task vectors (see Methods Targeted dimensionality reduction for details). The spiking activity of PFC neurons was shown to encode the reversal of behavior in a previous study (*Bartolo and Averbeck, 2020*). Following this line of work, we focused on analyzing the trials around the behavioral reversal point in each block. We referenced the position of each trial to the behavioral reversal trial as a relative trial.

Within each trial, the block-averaged neural activity associated with choices and inferred reversal probability, denoted as $x_{choice}$ and $x_{rev}$, respectively, produced non-stationary dynamics (*Figure 2A*, left). Their activity level reached a maximum around the time of cue onset (white squares in *Figure 2A*, left), when the monkey and RNN were about to make a choice. Such rotational neural dynamics were found both in the PFC of monkeys and the RNNs.

Across trials, the orientation of rotational trajectories shifted, indicating systematic changes in the choice and reversal probability activity. When the task-relevant activity at cue onset (or fixation) was analyzed, the points in the two-dimensional phase space ($x_{rev}$ and $x_{choice}$) shifted gradually across trials

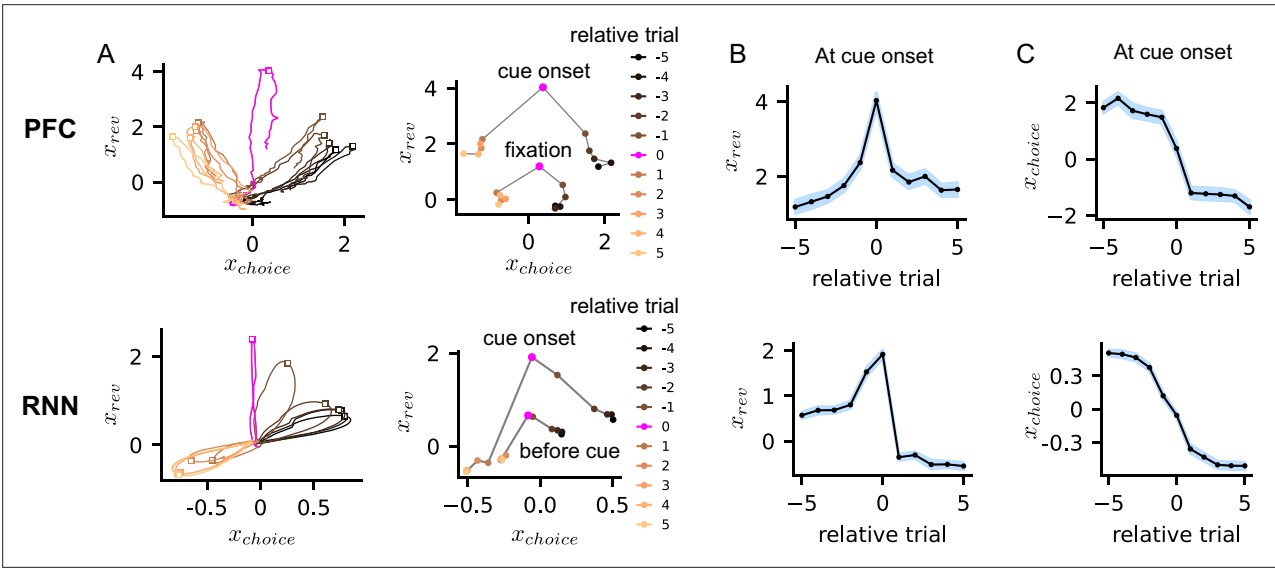

**Figure 2.** Neural trajectories encoding choice and reversal probability variables. (**A**) Neural trajectories of prefrontal cortex (PFC) (top) and RNN (bottom) obtained by projecting population activity onto task vectors encoding choice and reversal probability. Trial numbers indicate their relative position to the behavioral reversal trial. Neural trajectories in each trial were averaged over eight experiment sessions and 23 blocks for the PFC, and 40 networks and 20 blocks for the recurrent neural networks (RNNs). Black square indicates the time of cue onset. (**B–C**) Neural activity encoding reversal probability and choice in PFC (top) and RNN (bottom) at the time of cue onset (black squares in panel A) around the behavioral reversal trial. Shaded region shows the S.E.M over sessions (or networks) and blocks.

(*Figure 2A*, right). We found that reversal probability activity, $x_{rev}$, peaked at the reversal trial in the PFC and the RNN (*Figure 2B*). Choice activity, $x_{choice}$, on the other hand, decreased gradually over trials reflecting the changes in choice preference (*Figure 2C*). The inverted-V shape of $x_{rev}$ and the monotonic decrease of $x_{choice}$ over trials explained the counter-clockwise shift in the rotational trajectories observed in the two-dimensional phase space (*Figure 2A*).

A recent study found that, when a neural network was trained via reinforcement learning to perform a reversal learning task, the first two principal components of the network activity shifted gradually following a shape similar to $x_{rev}$ and $x_{choice}$ (see Figure 1 in *Wang et al., 2018*). These results suggest that the gradual shift in network states across trials (*Figure 2B,C*) could be a common feature that emerges in networks performing a reversal learning task, regardless of training methods. One main difference is that, by design, these neural networks lacked within-trial dynamics in contrast to our RNNs (*Wang et al., 2018*). In the following sections, we characterize the dynamics and properties of the non-stationary neural activity that our RNNs and the PFC of monkeys generated during the trials (*Figure 2A*).

## Integration of reward outcomes drives reversal probability activity

We first asked if there were underlying dynamics that systematically changed the reversal probability activity, $x_{rev}$. Previous works have shown that accumulation of decision-related evidence can be represented as a line attractor in a stable subspace of network activity (*Mante et al., 2013*; *Nair et al., 2023*). We hypothesized that the gradual shift of $x_{rev}$ (*Figure 2A, B*) could be characterized similarly by a line attractor model, where $x_{rev}$ is explained by integrating reward outcomes across trials.

To test this idea, we set up a reward integration equation $x_{rev}^{k+1}(t) = x_{rev}^{k}(t) + R_{\pm}^{k}(t)$ that predicts the next trial's reversal probability activity at time $t$ based on the current trial's reversal probability activity at time $t$ and the reward outcome, therefore, predicting across-trial reversal probability by integrating reward outcomes. Here, $t$ is a time point within a trial (e.g. $t = t_{cue}$ is the time of cue onset), $x_{rev}^{k}(t)$ is the reversal probability activity at $t$ on trial $k$, and $R_{\pm}^{k}(t)$ is an estimate of the shift in reversal probability activity at $t$ driven by trial $k$'s reward outcome (+ if rewarded and - if not rewarded. See Methods Reward integration equation for details).

When the reward integration activity at cue onset ($t_{cue}$) was analyzed, the predicted reversal probability activity was in good agreement with the actual reversal probability activity of PFC and RNN (example blocks shown in *Figure 3A,C*; prediction accuracy of all blocks shown in *Figure 3E*). In addition, we found that $x_{rev}^{k}(t_{cue})$ responded to reward outcomes consistently with how reversal probability itself would be updated: no reward increased the reversal probability activity in the next trial (*Figure 3B,D*; no reward), while a reward decreased it (*Figure 3B, D*; reward). At the behavioral reversal trial ($k = 0$), however, the reversal probability activity in the following trial ($k = 1$) decreased regardless of the reward outcome at the reversal trial.

When the reward integration equation was fitted to other time points in the trial (i.e. estimate $R_{\pm}^{k}(t)$ at other $t$ in the trial), the prediction accuracy remained stable over the trial duration (*Figure 3F*). This suggested that a line attractor model might be applicable throughout the trial. However, the reward integration equation is an algebraic relation and does not capture the dynamics of neural activity, such as the non-stationary activity during the trial (e.g. *Figure 2*). This observation led us to characterize the dynamics of reversal probability activity to assess whether it was compatible with the line attractor model.

## Augmented attractor model of reversal probability activity

We showed that the reversal probability activity encodes the accumulation of reward outcomes, which resembles a line attractor model (*Figure 3*). However, a direct application of the line attractor dynamics would imply that, when no decision-related evidence is presented within a trial, the reversal probability activity should remain stationary (*Figure 4A*, Stationary), which was incompatible with the non-stationary reversal probability activity observed during a trial (*Figure 2A*).

To better characterize the dynamics of reversal probability activity, we analyzed how its derivative $dx_{rev}/dt$ changes throughout a trial. In the PFC of monkeys, we found that the reversal probability activity was stationary at the start of a trial: $dx_{rev}/dt$ was close to zero before and during the fixation (*Figure 4B*, Stationary). However, the non-stationary activity emerged at cue onset and was associated with task-related behavior, such as making a choice and receiving reward. Specifically, $dx_{rev}/dt$

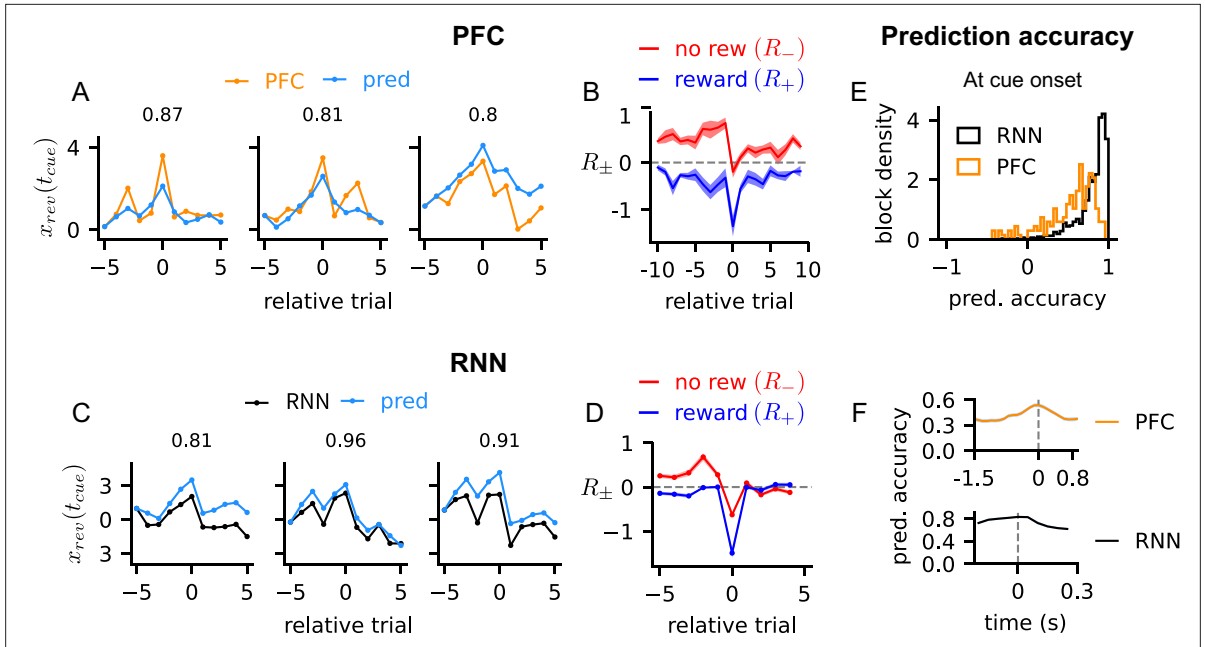

**Figure 3.** Integration of reward outcomes drives reversal probability activity. (**A**) The reversal probability activity of prefrontal cortex (PFC) (orange) and prediction by the reward integration equation (blue) at the time of cue onset across trials around the behavioral reversal trial. Three example blocks are shown. Pearson correlation between the actual and predicted PFC activity is shown on each panel. Relative trial number indicate the trial position relative to the behavioral reversal trial. (**B**) $R_{\pm}^k$ of PFC estimated from the reward integration equation at cue onset. $R_{-}^k$ and $R_{+}^k$ correspond to no-reward (red) and reward trials (blue), respectively. The shaded region shows the S.E.M over blocks and sessions. (**C–D**) Same as in panels (**A**) and (**B**) but for trained recurrent neural networks (RNNs). (**E**) Prediction accuracy, quantified with Pearson correlation, of the reward integration equation of all eight PFC recording sessions and all 40 trained RNNs at cue onset. (**F**) Average prediction accuracy, quantified with Pearson correlation, of the reward integration equation across time. The value at each time point shows the prediction accuracy averaged over all blocks in PFC recording sessions (top) or trained RNNs (bottom).

increased rapidly at the cue onset, when a decision was made, followed by a sharp fluctuation and slow decay until the reward time (*Figure 4B*, Non-stationary).

We then analyzed the contraction factor of the reversal probability activity to assess whether the activity is contracting or expanding around its mean activity (*Figure 4C*; see Methods Contraction factor of reversal probability activity for details). We found that the contraction factor was less than 1 (i.e. contracting) before the fixation period (-2.5 to -1 s in *Figure 4C*, left; also see the right panel), became close to 1 (i.e. marginally stable) around fixation (-1 to 0 s), and exceeded 1 (i.e. expanding) at cue onset (0 s). This showed that the reversal probability activity is a point attractor (i.e. stationary and contracting) at the start of a trial but loses its stability as task-related behavior is executed.

In the RNNs, we found that the dynamics of reversal probability activity were similar to the PFC's activity when an additional constraint was added to the RNN's objective function to fix its choice output to zero before making a choice (i.e. fixation period was between fixation and cue-off lines in *Figure 4D*; indicated as Stationary period). With the fixation constraint, the reversal probability activity was stationary during the fixation period (i.e. $dx_{rev}/dt$ converged and stayed close to zero). This contrasted the RNNs trained without the fixation, which exhibited more dynamics before making a choice, suggesting the role of fixation in promoting stationary dynamics (*Figure 4—figure supplement 1*). The RNN's non-stationary dynamics after cue-off was consistent with the PFC's non-stationary activity regardless of the fixation constraint (*Figure 4* and *Figure 4—figure supplement 1*, Non-stationary).

The contraction factor of reversal probability activity of the RNNs trained with the fixation constraint showed a trend that was similar to the PFC. It was less than 1 during fixation (fixation to cue-off in *Figure 4E*; also see the right panel) and became close to 1 immediately after cue-off when the decision was made (the point next to cue-off in *Figure 4E*). As in the PFC, the contraction factor shows that the RNN's reversal probability activity is a point attractor at the start of a trial and becomes unstable as the RNN executes decision making.

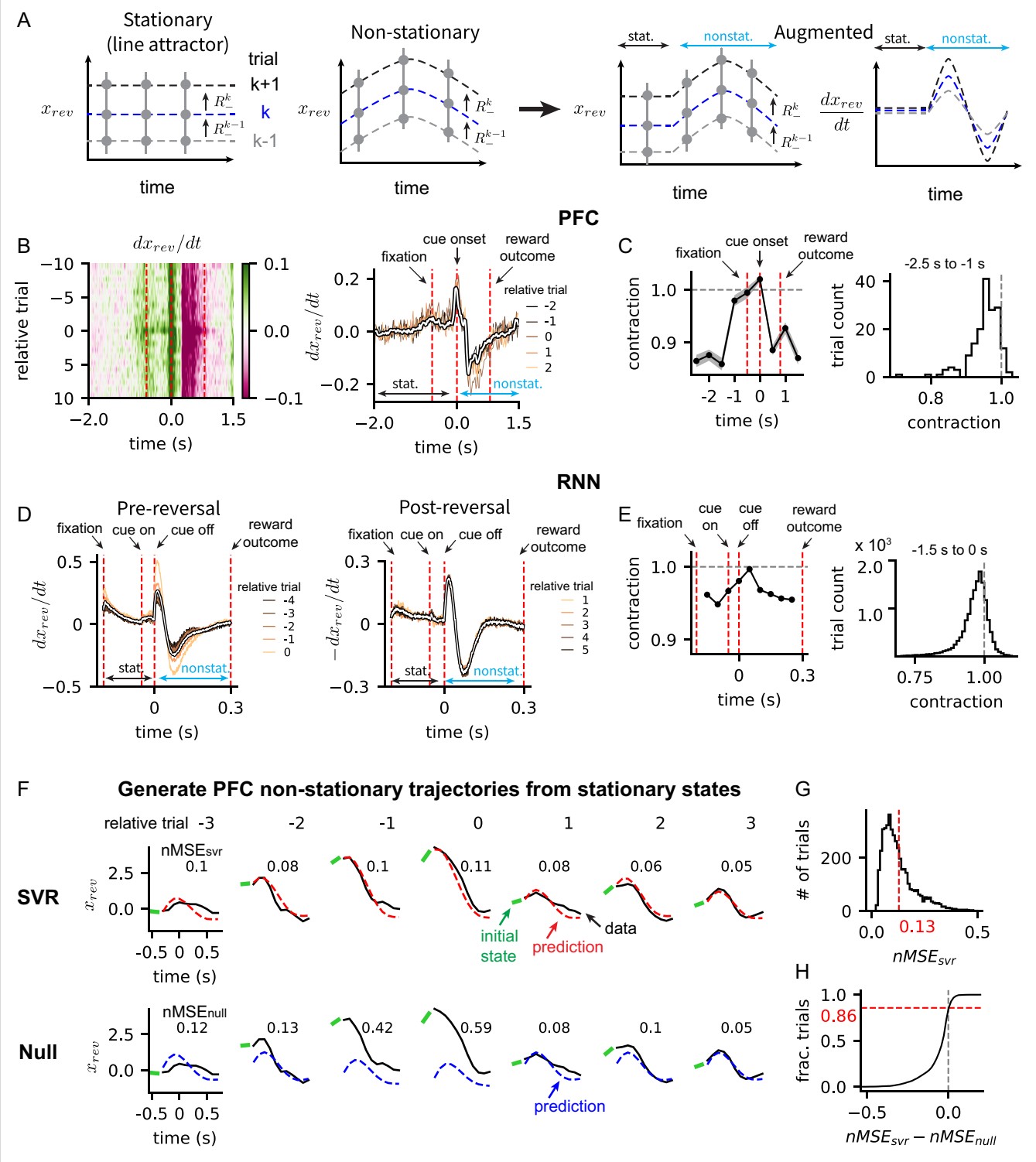

**Figure 4.** Augmented model for reversal probability activity. (**A**) Schematic of two activity modes of the reversal probability activity. Left: Stationary mode (line attractor) where $x_{rev}(t)$ remains constant during a trial, and non-stationary mode where $x_{rev}(t)$ is dynamic. Right: Augmentation of stationary and non-stationary activity modes where the stationary mode leads the non-stationary mode in time. The time derivative $dx_{rev}/dt$ is shown to demonstrate (non-)stationarity of the activity. (**B**) Left: Block-averaged $x_{rev}/dt$ of prefrontal cortex (PFC) across trial and time. Dotted red lines indicate the onset time of fixation (-0.5 s), cue (0 s) and reward (0.8 s); same lines shown on the right. Right: $x_{rev}/dt$ averaged over all trials (white), together with the trajectories of five trials around the reversal trial (colored). (**C**) Left: Contraction factor of $x_{rev}$ of PFC at different time points. Dotted line at 1 indicates the threshold of contraction and expansion. Right: Contraction factor of PFC $x_{rev}$ of individual trials between the time interval -2.5 and -1

*Figure 4 continued on next page*

*Figure 4 continued*

s. (**D**) Block-averaged $dx_{rev}/dt$ of recurrent neural networks (RNNs) at the pre-reversal (left) and post-reversal (right) trials. Note that the sign of the post-reversal trial trajectories was flipped to match the shape of the pre-reversal trajectories. Dotted red lines indicate the time of fixation, cue off, and reward. (**E**) Contraction factor of $x_{rev}$ of RNN. Similar results for RNN as in panel (**C**). (**F**) Generating PFC non-stationary reversal probability trajectories from the stationary activity using support vector regression (SVR) models. Top: Trajectories generated from SVR compared to the PFC reversal probability trajectories in trials around the reversal trial in an example block. The initial state (green) is the input to the SVR model, which then predicts the rest of the trajectory. The normalized mean-squared-error (MSE) between the SVR trajectory (prediction, red) and the PFC trajectory (data, black) is shown in each trial. Bottom: Trajectories generated from the null SVR compared to the PFC reversal probability trajectories. The initial states of trials in a block were shuffled randomly prior to training the null SVR model. The trajectories predicted from the null SVR model (blue) are compared to the PFC reversal probability trajectories (black). (**G**) The normalized MSE of all trials in the test dataset. (**H**) Difference between the normalized MSE of the SVR and the null models. The difference in normalized MSE between two models was calculated for each trial.

The online version of this article includes the following figure supplement(s) for figure 4:

**Figure supplement 1.** Comparison of recurrent neural networks (RNNs) trained with and without fixation.

Thus, analyzing $dx_{rev}/dt$ across a trial showed that the reversal probability activity consisted of two dynamic modes: a point attractor at the start of a trial, which was consistent with the line attractor model, followed by non-stationary dynamics during the trial, which deviated from the line attractor.

Next, we investigated whether the two activity modes are linked by common dynamics. In the RNNs, the cue applied at the end of the stationary period (*Figure 4D*, cue-on to cue-off) determined the initial state from which the non-stationary activity was generated by the recurrent dynamics. We wondered if the two activity modes of PFC also obeyed the same dynamic relationship, i.e., the non-stationary activity is generated by underlying dynamics with an initial condition given by the stationary state.

To test this hypothesis, we took the PFC activity at the start of fixation period (-0.5 s) as the initial state. Then, we trained a support vector regression (SVR) to infer the underlying dynamics, which used the PFC activity at fixation as the input and generated the remaining trajectory until the reward time as the output. The SVR model was trained on neural data from 20 trials around the behavioral reversal in 10 randomly selected blocks from a session and tested on the remaining blocks (approximately 10 blocks) from the same session, thus training separate SVR model for each session. This procedure was repeated 10 times to sample different sets of training blocks (see Methods Generating the PFC reversal probability trajectories from initial states for the details). The prediction error was quantified using the normalized mean-squared-error (MSE) between the SVR prediction and the actual reversal probability trajectory normalized by the amplitude of the trajectory.

*Figure 4F* (top) compares the SVR trajectory (red, prediction) generated from an initial state (green, initial state) and the actual PFC reversal probability trajectory (black, data) in trials around the behavioral reversal. We found that the SVR trajectories were able to capture the overall shape of data trajectories that had a bump shape and shifted up and down with the initial state. The normalized MSE of all the trials in the test dataset was 0.13, i.e., the mean error is 13% of the trajectory amplitude (*Figure 4G*).

To verify the role of initial state in generating trajectories, we trained a null SVR model, in which the initial states of trials in a block were randomly shuffled before training the model. In other words, a reversal probability trajectory was generated not from its own initial state, but from the initial state of a randomly chosen trial. The rest of the training procedure was identical as described above. We found that, although the null trajectory (*Figure 4F*, Null, prediction) resembled the overall shape of the data trajectory (*Figure 4F*, Null, data), i.e., increase towards cue onset and then decrease monotonically, it did not shift together with the initial states (*Figure 4F*, Null, initial state), as did the data trajectories. When the normalized MSE of the SVR and the null trajectories were compared, we found that in 86% of the test trials the SVR error was smaller than the null error (*Figure 4H*).

Together, our results show that the reversal probability activity consists of two activity modes linked by underlying dynamics. The stable stationary state (i.e. point attractor) at the start of a trial determines the initial condition from which the non-stationary dynamics, associated with task-related behavior, is generated. These findings suggest an extension of the standard line attractor model, in which points on the line attractor serve as initial states that launch non-stationary activity necessary to perform a task.

## Dynamic neural trajectories encoding reversal probability are separable

The previous section showed that the underlying neural dynamics, associated with task-related behavior, generates non-stationary activity during a trial. We next asked how distinct probabilistic values can be encoded in this non-stationary activity. In a stationary state, different positions encode different levels of decision-related evidence. When stationary points evolve through the non-stationary dynamics, as in our augmented model, they must remain separable in order to represent distinct values.

To address this question, we compared trajectories of adjacent trials to examine if the reward outcome drives the next trial's trajectory away from the current trial's trajectory, thus separating

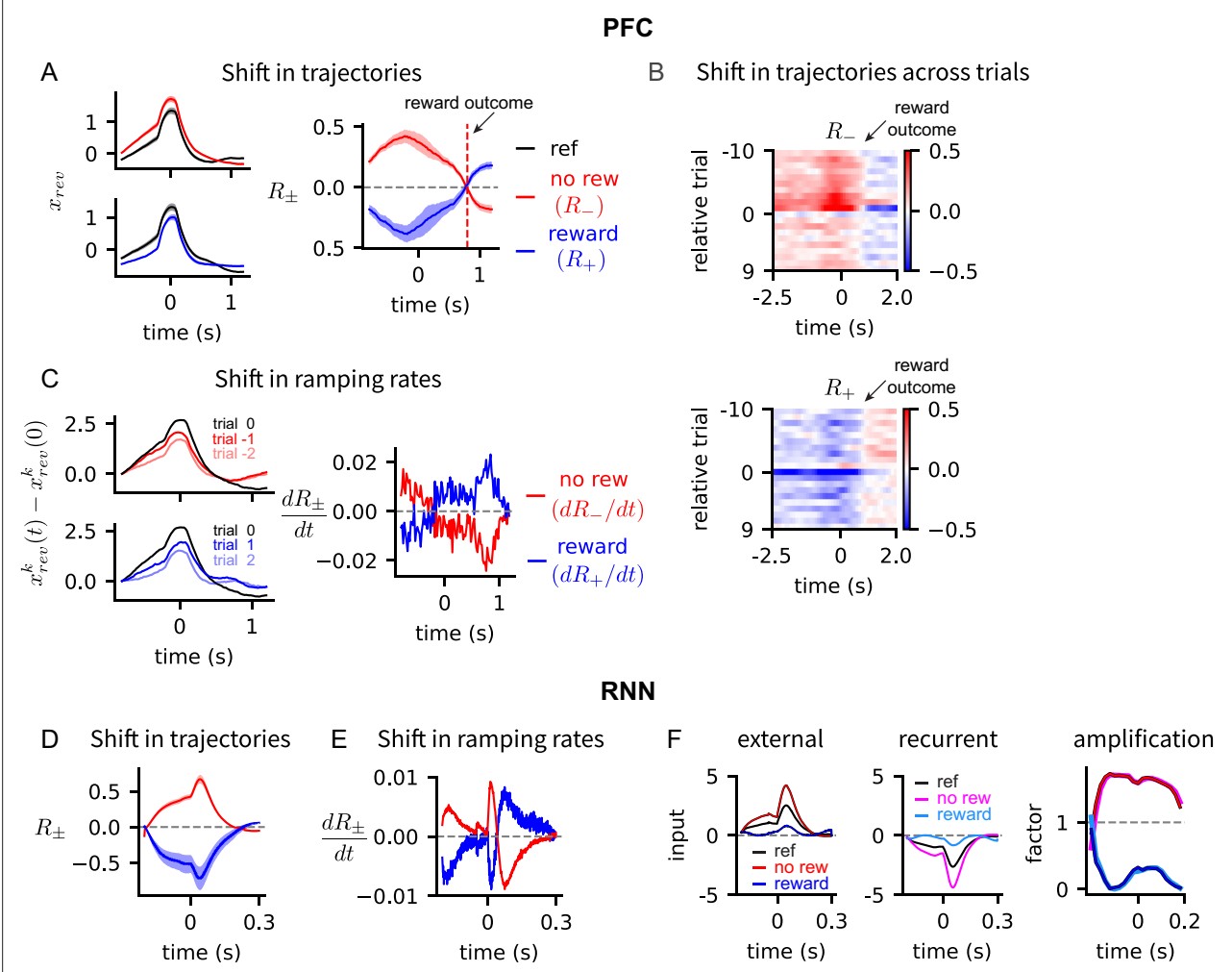

**Figure 5.** Dynamic neural trajectories encoding reversal probability are separated in response to reward outcomes. (**A**) Left: $x_{rev}(t)$ of prefrontal cortex (PFC) at current trial (black) is compared to $x_{rev}(t)$ in the next trial when reward is received (top, red) and not received (bottom, blue). Right: The difference of $x_{rev}(t)$ between current and next trials shown on the left panels. Shaded region shows the S.E.M. across all trials, blocks, and sessions. (**B**) Difference of $x_{rev}$ of two adjacent trials when reward is not received (top, $R_-$) or received (bottom, $R_+$). The approximate time of reward outcome is shown. Relative trial number indicate the trial position relative to the behavioral reversal trial. (**C**) Left: $x_{rev}(t)$ of PFC of consecutive no reward trials before the behavioral reversal trial (top) and consecutive reward trials after the behavioral reversal (bottom). The initial value was subtracted to compare the ramping rates of $x_{rev}(t)$. Right: Difference in the ramping rates of trajectories of adjacent trials, when reward was received (blue) and not received (red). (**D–E**) Same as the right panels in (**A**) and (**C**) but for trained recurrent neural networks (RNNs). (**F**) Left, Middle: External (left) and recurrent (middle) inputs to the RNN reversal probability dynamics, when reward was not received (red, magenta) or was received (blue, cyan). Right: Amplification factor shows the ratio of the total input when no reward (or reward) was received to the total input of reference input. The amplification factors for both the external (red, blue) and recurrent (magenta, cyan) inputs are shown. Red and magenta curves and blue and cyan curves overlap.

The online version of this article includes the following figure supplement(s) for figure 5:

**Figure supplement 1.** Decoding reward outcome and the behavioral reversal trial using neural trajectories encoding reversal probability.

them. Analysis of PFC activity showed that not receiving a reward increased the next trial's trajectory $x_{rev}^{k+1}(t)$, compared to the current trial's trajectory $x_{rev}^{k}(t)$, over the entire trial duration until the next trial's reward was revealed (*Figure 5A*, left; $R\_$). This was shown in the difference of adjacent trials' trajectories being positive values, when not rewarded through most of the trial (*Figure 5A*, right; $R\_$). Moreover, across trials, the same trend was observed in all the trials except at the behavioral reversal trial, at which the reversal probability activity reached its maximum value and decreased in the following trial (*Figure 5B*, top; $R\_$). On the other hand, when a reward was received, the next trial's trajectory was decreased, compared to the current trial's trajectory over the entire trial duration until the next trial's reward (*Figure 5A*, left; $R_+$). This was shown in the difference of adjacent trials' trajectories being negative values, when rewarded through most of the trial (*Figure 5A*, right; $R_+$). Across trials, the same trend was observed in all the trials except at the trial preceding the behavioral reversal trial, at which the trajectory increased to the maximum value at the reversal trial (*Figure 5*, bottom; $R_+$). Additional analysis on $R\_$ and $R_+$ beyond the next trial's reward time can be found in *Figure 6—figure supplement 1*.

We then examined what type of activity the dynamic trajectories exhibited when separating away from the previous trial's trajectory. Ramping activity is often observed in cortical neurons of animals engaged in decision-making (*Li et al., 2015*; *Li et al., 2016*; *Finkelstein et al., 2021*; *Latimer et al., 2015*; *Zoltowski et al., 2019*). We found that when no rewards were received, trajectories were separated from the previous trial's trajectory by increasing their ramping rates towards the decision time ($dR\_/dt > 0$ in *Figure 5C*, right). On the other hand, when rewards were received, trajectories were separated by decreasing their ramping rate ($dR_+/dt < 0$ in *Figure 5C*, right). The increase (or decrease) in the ramping rates was observed in consecutive no reward (or reward) trials around the reversal trial (*Figure 5C*, left).

Consistent with the PFC activity, the trained RNN exhibited similar activity responses to reward outcomes: neural trajectories encoding reversal probability increased, when not rewarded, and decreased, when rewarded. The shift in trajectories persisted throughout the trial duration (*Figure 5D*) and ramping rates changed in agreement with the PFC findings (*Figure 5E*).

We examined the circuit dynamic motif of the trained RNN that separates neural trajectories. We projected the differential equation governing the network dynamics onto a one-dimensional subspace encoding reversal probability and analyzed the contribution of external and recurrent inputs to reversal probability dynamics: $\tau \dot{x}_{rev} = x_{rec} + x_{ext} \equiv x_{total}$ (see Methods Recurrent neural network for details). We found that the effect of the external input $x_{ext}$ was positive, reflecting that the external feedback input drives the reversal probability. On the other hand, the recurrent input $x_{rec}$ was negative, showing that it curtailed the external input (*Figure 5F*, external and recurrent). To analyze the effects of no-reward (or reward), we averaged the reversal probability activity over all trials (Fig.5F, reference) at which the subsequent trial was not (or was) rewarded. When no reward was received, $x_{ext}$ (*Figure 5F*, external, red) and $x_{rec}$ (*Figure 5F*, recurrent, magenta) were both amplified, compared to the reference, by approximately the same factor and resulted in increased total input: $x_{total}^{no\ rew} = \gamma^{no\ rew} x_{total}$ with $\gamma^{no\ rew} > 1$ (*Figure 5F*, amplification, red and magenta). On the other hand, when reward was received, $x_{ext}$ (*Figure 5F*, external, blue) and $x_{rec}$ (*Figure 5F*, recurrent, cyan) were both suppressed, resulting in decreased total input: $x_{total}^{reward} = \gamma^{reward} x_{total}$ with $\gamma^{reward} < 1$ (*Figure 5F*, amplification, blue and cyan). This suggests a circuit dynamic motif, where positive external feedback balanced by recurrent inhibition drives the reversal probability dynamics. The total drive is amplified or suppressed, depending on reward outcomes, resulting in a trajectory that separates from the previous trial's trajectory.

In sum, our findings show that dynamic neural trajectories encoding reversal probability are separated from the previous trial's trajectory in response to reward outcomes, allowing them to represent distinct values of reversal probability during a trial.

## Separability of reversal probability trajectories across trials

We investigated if reversal probability trajectories across multiple trials maintained separability. To this end, we analyzed the mean behavior of trajectories in each trial (referred to as mean trajectory of a trial) and analyzed their separability across trials around the behavioral reversal. Since a mean trajectory of a trial was obtained by averaging over all reward outcomes of the previous trial, we compared how reward and no-reward contributed to modifying the next trial's mean trajectory, which were quantified by $R_+^k$ and $R\_^k$ in *Figure 5B*, respectively.

We found that the effect of no-reward was larger than the effect of reward before the behavioral reversal. This is shown as the trace $R_-^k$ (no reward) lying above the trace $-R_+^k$ (reward) during a trial and across all pre-reversal trials (see pre-reversal trials -5 to -1 in *Figure 6A*; $R_-^k$ and $-R_+^k$ are positive traces since $R_+^k$ is mostly negative). The temporal averages of $R_-^k$ and $-R_+^k$ captured this systematic differences in the pre-reversal trials (*Figure 6B*, bottom), and the sum $R_-^k + R_+^k$ was positive during a trial in 80% of the pre-reversal trials (*Figure 6C*, top). The positivity of $R_-^k + R_+^k$ in time and across pre-reversal trials implied that the mean trajectories remained separable by increasing monotonically across the trials (*Figure 6C*, bottom). Consistent with this observation, the Spearman rank correlation of pre-reversal trajectories was stable in time (*Figure 6E*, pre).

After the behavioral reversal, the effects of no-reward and reward were the opposite of pre-reversal trials. The trace $-R_+^k$ lied above the trace $R_-^k$ (see post-reversal trials 0 to 4 in *Figure 6A*), and the temporal average $\langle -R_+^k \rangle_t$ was larger than $\langle R_-^k \rangle_t$ (*Figure 6B*, bottom). This showed that $R_-^k + R_+^k$ was mostly negative during post-reversal trials. The fraction of trials, for which $R_-^k + R_+^k$ is negative, was close to 80% among the post-reversal trials (*Figure 6D*, top). The negativity of $R_-^k + R_+^k$ in time and across post-reversal trials implied that the post-reversal trajectories remained separable by decreasing monotonically across the trials (*Figure 6D*, bottom). The order of post-reversal trajectories was stable over the trial duration, similarly to the pre-reversal trials but in the reversed order (*Figure 6E*, post).

In the trained RNNs, the effects of reward outcomes on mean trajectories were consistent with the PFC findings: $R_-^k + R_+^k$ was positive and negative before and after the behavioral reversal, respectively (*Figure 6F*). Consequently, reversal probability trajectories of the RNNs maintained separability by shifting monotonically as in the PFC (*Figure 6G,H*). The order of trajectories was also stable over the trial duration (*Figure 6I*).

These findings show that the mean behavior of reversal probability trajectories is to shift monotonically across trials. It suggests that a family of separable neural trajectories around the reversal trial can represent varying estimates of reversal probability stably in time.

## Perturbing reversal probability activity biases choice outcomes

Here, we turned to the RNN to see if we could perturb activity within the reversal probability space and consequently perturb the network's choice preference. We defined the reversal probability activity $x_{rev}$ as the activity in a neural subspace correlated to the behavioral reversal probability (Methods Targeted dimensionality reduction). However, it remains to be shown if the activity within this neural subspace can causally affect network's behavioral outcomes.

Previous experimental works demonstrated that perturbing neural activity of medial frontal cortex (*Murphy et al., 2022*), specific cell types (*Yi et al., 2024*; *Jeong et al., 2020*), or neuromodulators (*Su and Cohen, 2022*; *Hyun et al., 2023*) affect the performance of reversal learning. In our perturbation experiments in the RNNs, the perturbation was tailored to be within the reversal probability space by applying an external stimulus aligned ($v_+$) or opposite ($v_-$) to the reversal probability vector. An external stimulus in a random direction was also applied as a control ($v_{rnd}$). All the stimuli were applied before the time of choice at the reversal trial or at preceding trials (*Figure 7*).

We found that the deviation of perturbed reversal probability activity from the unperturbed activity peaked at the end of perturbation duration and decayed gradually (*Figure 7B*, red solid). The perturbed choice activity, however, deviated more slowly and peaked during the choice duration (*Figure 7B*, black solid). This showed that perturbation of the reversal probability activity had its maximal effect on the choice activity when the choice was made. The strong perturbative effects on the reversal probability and choice activity were not observed in the control (*Figure 7B*, dotted).

The perturbation in the aligned ($v_+$) and opposite ($v_-$) directions shifted the reversal probability activity along the same direction as the perturbation vector, as expected (*Figure 7C*, left). On the other hand, the choice activity was increased when the perturbation was in the opposite direction $v_-$. The perturbation in the aligned direction $v_+$ did not decreases the choice activity substantially, but its effect was significantly smaller than the increase seen in $v_-$ perturbation (*Figure 7C*, right; KS-test, p-value = 0.007).

We further analyzed if perturbing within the reversal probability space could affect the choice outcomes, specifically the behavioral reversal trial. We found that the reversal trial was delayed when $v_-$ stimulus was applied to reduce the reversal probability activity (*Figure 7D*, left). The effect of $v_-$ stimulus increased gradually with the stimulus strength and was significantly stronger than the $v_+$ or

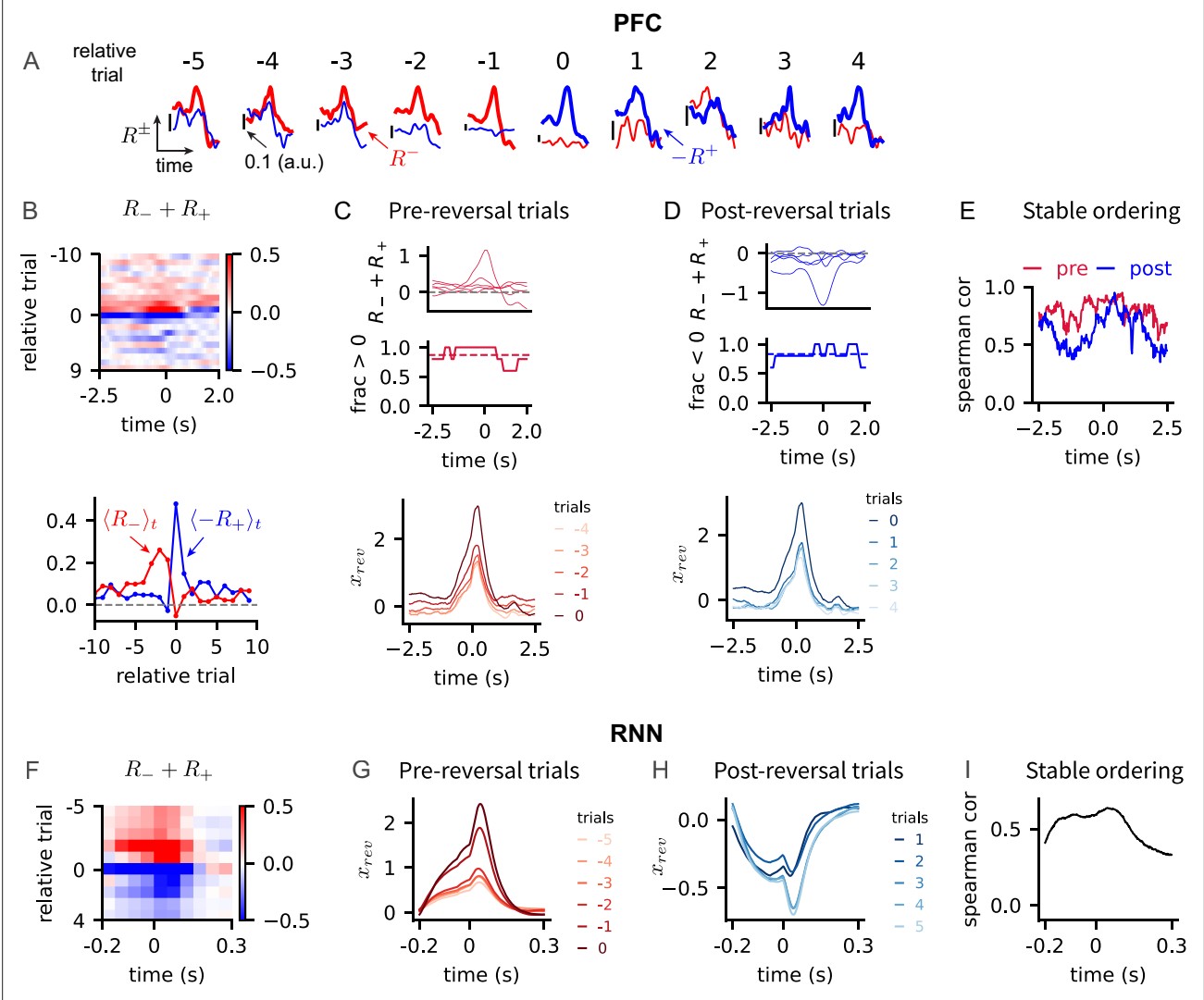

**Figure 6.** Mean trajectories encoding reversal probability shift monotonically across trials. (**A**) Traces of $R_-^k$ and $-R_+^k$ around the behavioral reversal trial. Note the sign flip in $-R_+^k$, which was introduced to compare the magnitudes of $R_-^k$ and $R_+^k$. Relative trial number indicate the trial position relative to the behavioral reversal trial. (**B**) Top: $R_-^k + R_+^k$ across trial and time. Bottom: Temporal averages of $R_-^k$ and $-R_+^k$ over the trial duration. (**C**) Top: Traces of $R_-^k + R_+^k$ of pre-reversal trials (relative trial $k = -5$ to -1), and the fraction of trials at each time point that satisfy $R_-^k + R_+^k > 0$. Bottom: Mean prefrontal cortex (PFC) reversal probability trajectories of pre-reversal trials. (**D**) Same as in panel (**C**), but for post-reversal trials (relative trial $k = 0$ to 4). (**E**) Spearman rank correlation between trial numbers and the mean PFC reversal probability trajectories across pre-reversal (red) and post-reversal (blue) trials at each time point. For the post-reversal trials, Spearman rank correlation was calculated with the trial numbers in reversed order to capture the descending order. (**F**) $R_-^k + R_+^k$ of trained recurrent neural networks (RNNs) across trial and time. (**G**–**I**) Trained RNNs' block-averaged $x_{rev}$ before and after the reversal trial and their average Spearman correlation at each time point.

The online version of this article includes the following figure supplement(s) for figure 6:

**Figure supplement 1.** Break down of $R^+, R^-$ by the reward outcomes of two consecutive trials.

$v_{rnd}$ stimuli in delaying the reversal trial. Perturbation had the strongest effect when applied to the reversal trial, while perturbations on trials preceding the reversal showed appreciable but reduced effects (*Figure 7D*, right). When the $v_+$ stimulus was applied to trials preceding the reversal trial, the reversal was accelerated (*Figure 7E*, left). The effect of $v_+$ stimulus also increased with stimulus strength and was significantly stronger than the $v_-$ or $v_{rnd}$ stimuli in accelerating the reversal trial (*Figure 7E*, right).

We asked if perturbation of neural activity in PFC could exhibit similar responses. In other words, does increase (or decrease) in reversal probability activity lead to decrease (or increase) in choice

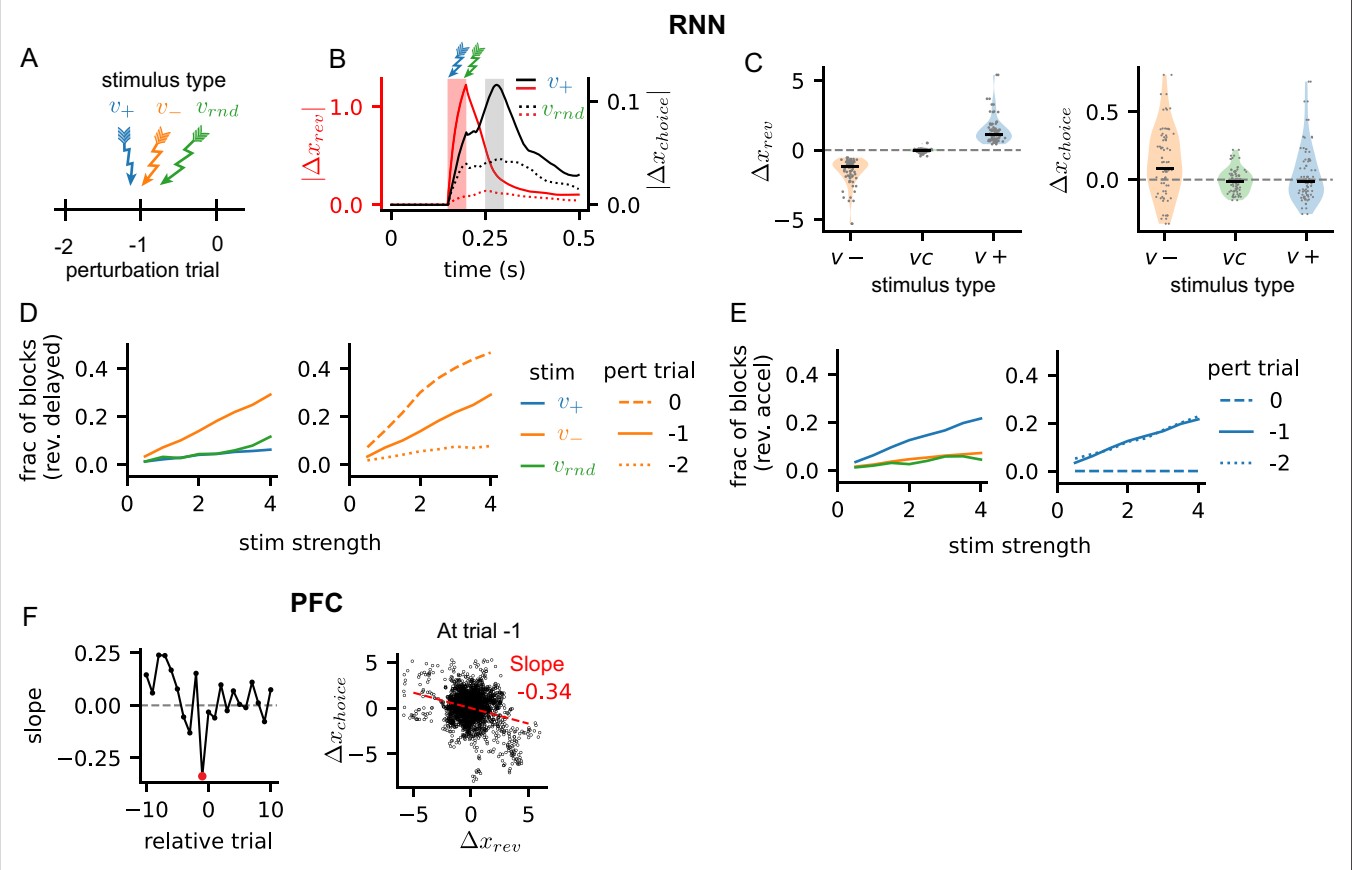

**Figure 7.** Perturbing recurrent neural networks (RNN's) neural activity encoding reversal probability biases choice outcomes. (**A**) RNN perturbation scheme. Three perturbation stimuli were used; $v_+$, population vector encoding the reversal probability; $v_-$, negative of $v_+$; $v_{rnd}$, control stimulus in random direction. Perturbation stimuli were applied at the reversal (0) and two preceding (-2, -1) trials. (B) Deviation of reversal probability activity $\Delta x_{rev}$ and choice activity $\Delta x_{choice}$ from the unperturbed activity. Perturbation was applied at the reversal trial during a time interval the cue was presented (shaded red). Choice was made after a short delay (shaded gray). Perturbation response along the reversal probability vector $v_+$ (solid) and random vector $v_{rnd}$ (dotted) are shown. (**C**) Perturbation of reversal probability activity (left) and choice activity (right) in response to three types of perturbation stimuli. Each dot shows the response of a perturbed network. Two perturbation strengths (multiplicative factor of 3 and 4 shown in panels D and E) were applied to 40 RNNs. $\Delta x_{rev}$ shows the activity averaged over the duration of perturbation, and $\Delta x_{choice}$ shows the averaged activity over the duration of choice. $\Delta x_{choice}$ of $v_+$ is significantly smaller than $\Delta x_{choice}$ of (KS-test, p-value = 0.007). (**D–E**) Fraction of blocks in all 40 trained RNNs that exhibited delayed or accelerated reversal trials in response to perturbations of the reversal probability activity. Perturbations at trial number -1 by three stimulus types are shown on the left panels, and perturbations at all three trials by the stimulus of interest ($v_-$ in **D** and $v_+$ in **E**) are shown on the right panels. A multiplicative factor on the perturbation stimuli is shown as stimulus strength. (F) Left: The slope of linear regression model fitted to the residual activity of reversal probability and choice. The residual activity at each trial over the time interval [0, 500]ms was used to fit the linear model. Red dot indicates the slope at trial number -1. Relative trial number indicate the trial position relative to the behavioral reversal trial. Right: Each dot is the residual activity of a block at trial number -1. Red line shows the fitted linear model, and its slope (-0.34) is shown.

activity in PFC? Although PFC activity was not perturbed by external inputs, we considered the residual activity of single trials, i.e. deviation of single trial neural activity around the trial-averaged activity, to be 'natural' perturbation responses. We fitted a linear model to the residual activity of reversal probability and choice and found that they were strongly negatively correlated (i.e. negative slope in *Figure 7F*) at the trial preceding the behavioral reversal trial. This analysis demonstrated the correlation between perturbation responses of reversal probability and choice activity. However, it remains to be investigated, through perturbation experiments, whether reversal probability activity is causally linked to choice activity in PFC and, moreover, to animal's choice outcomes.

## Discussion

### Reversal learning

Reversal learning has been a behavioral framework for investigating how the brain supports flexible behavior (*Butter, 1969*; *Costa et al., 2015*; *Groman et al., 2019*; *Bartolo and Averbeck, 2020*; *Su and Cohen, 2022*; *Hyun et al., 2023*) and for elucidating neural mechanisms underlying mental health issues (*Groman et al., 2022*; *Schoenbaum et al., 2006*). It has been shown that multiple brain regions (cortical *Groman et al., 2019*; *Bartolo and Averbeck, 2020*; *Wilson et al., 2010*; *Murphy et al., 2022*; *Yi et al., 2024*; *Jeong et al., 2020*; *Wilson et al., 2014* and subcortical *Groman et al., 2019*; *Averbeck and O'Doherty, 2022*), neuromodulators ; (*Costa et al., 2015*; *Su and Cohen, 2022*; *Hyun et al., 2023*), and different inhibitory neuron types (*Yi et al., 2024*; *Jeong et al., 2020*) are involved in reversal learning.

### Our results

Despite these recent advances, the dynamics of neural activity in cortical areas during a reversal learning task have not been well characterized. In this study, we investigated how reversal probability is represented in cortical neurons by analyzing neural activity in the prefrontal cortex of monkeys and recurrent neural networks performing the reversal learning task. We found that the activity in a neural subspace encoding reversal probability represented integration of reward outcomes. This reversal probability activity had two activity modes: stable stationary activity at the start of trial, followed by non-stationary activity during the trial. There was underlying dynamics, associated with task-related behavior, that generated the non-stationary activity with an initial condition given by the stationary state. The existence of two activity modes suggested an extension of the standard line attractor model where non-stationary dynamics driven by task-related behavior link the attractor states. The non-stationary trajectories were separable, allowing distinct probabilistic values to be represented in dynamic trajectories. Perturbation experiments in the RNNs demonstrated a potential causal link between reversal the probability activity and choice outcomes.

### Attractor dynamics

RNNs with attractor dynamics have been investigated in various contexts as a neural implementation of normative models of decision-making and evidence integration (*Ratcliff, 1978*; *Palmer et al., 2005*; *Shadlen and Newsome, 2001*; *Mazurek et al., 2003*; *Ratcliff et al., 2003*). One perspective is to consider decision variables as discrete or continuous attractor states of an RNN. Then, the network activity converges to an attracting state as a decision is made. Biologically plausible network models (*Wang, 2002*; *Seung, 1996*) and neural recordings in cortical areas have been shown to exhibit discrete (*Luo et al., 2023*; *Sutton, 1988*; *Genkin et al., 2023*) and continuous (*Wimmer et al., 2014*) attractor dynamics. Another perspective, more closely related to our study, is to consider evidence integration as a movement of network state along a one-dimensional continuous attractor, as demonstrated in *Inagaki et al., 2019*; *Nair et al., 2023*; *Sylwestrak et al., 2022* (see also continuous attractor dynamics in spatial mapping *Bollimunta et al., 2012*; *Gardner et al., 2022*; *Sorscher et al., 2023*; *Hulse and Jayaraman, 2020*).

In most of the studies, decision-related evidence was presented without significant interruption until the decision point (*Luo et al., 2023*; *Sutton, 1988*; *Mante et al., 2013*; *Genkin et al., 2023*). However, this was not the case in a reversal learning task with probabilistic rewards, as reward outcomes were revealed intermittently over multiple trials while task-related behavior must be performed within each trial. We showed that such multi-trial evidence integration promoted substantial non-stationary activity in the neural subspace encoding reversal probability. Therefore, the continuous attractor dynamics, in which the network state stays close to the attracting states, did not fully account for the observed neural dynamics. Instead, our findings suggest that separable dynamic trajectories in addition to attractor states could serve as a neural mechanism that represents accumulated evidence and accommodates non-stationary behaviors necessary to perform the task.

### Limitations

Our work demonstrated similarities in how reversal probability is represented in the PFC of monkeys and the RNNs. However, our approach, which compares the activity of RNNs trained on the task to the PFC activity, does not directly characterize the dynamic landscape of PFC activity. In particular,

our analysis only shows that the PFC activity at the initial state of each trial can be characterized as a point attractor (*Figure 4C, E*). Although this result is compatible with a line attractor model, it does not demonstrate whether the initial states across trials collectively form a line attractor. In order to show the existence of a line attractor, it is necessary to identify the dynamics in the region of the phase space occupied by the set of point attractors.

Alternative approaches that infer the latent dynamics of spiking activity of a neural population (*Kim et al., 2023*; *Kim et al., 2021*; *Chen et al., 2018*; *Sussillo et al., 2016*) could allow us to address this question by deriving a low-dimensional system of differential equations that generates the PFC activity. If the inferred latent dynamics could generate the two activity modes observed in our data, it would allows us to characterize different aspects of the neural dynamics, such as attractor states, the role of external inputs, and the non-stationary dynamics, within a single dynamical system model.

## Related work

Recent studies showed that intervening behaviors, such as introducing an intruder (*Nair et al., 2023*) or accumulating reward across trials (*Sylwestrak et al., 2022*), could produce neural trajectories that deviate from and retract to a line attractor. These studies are consistent with our finding in that their neural dynamics temporally deviated from attractor states. However, in our study, we did not thoroughly investigate the neural activity from dynamical systems perspective. It remains as future work to characterize the dynamical landscape of how the separable dynamic trajectories observed in our study are augmented to the continuous attractor model and compare it to the previous works (*Nair et al., 2023*; *Sylwestrak et al., 2022*).

A number of relevant studies have trained RNNs to perform various decision-making tasks (*Wang et al., 2018*; *Schaeffer et al., 2020*; *Song et al., 2017*; *Molano-Mazón et al., 2023*). In a related work (*Schaeffer et al., 2020*), RNNs were trained to perform a change point detection task designed by the International Brain Laboratory (*Findling, 2023*). They showed that trained RNNs exhibited behavior outputs consistent with an ideal Bayesian observer without explicitly learning from the Bayesian observer. This finding shows that the behavioral strategies of monkeys could emerge by simply learning to do the task, instead of directly mimicking the outputs of Bayesian observer as done in our study.

The trained RNN in their work (*Schaeffer et al., 2020*) exhibited line attractor dynamics in contrast to our observation of stationary and non-stationary dynamics (*Figure 4A*). In another study, a line attractor-like dynamics, where the principal components of network activity moved gradually across trials, was observed in artificial agents trained to perform the reversal learning task via reinforcement learning (seeFigure 1 in *Wang et al., 2018*). One possible reason for the lack of non-stationary dynamics within a trial in these other studies is that a trial consisted of only one (*Wang et al., 2018*) or a few time points (*Schaeffer et al., 2020*), which limits the possible range of temporal dynamics RNNs can exhibit during a trial. This suggests that different setup of a task can lead to significantly different dynamics in the trained RNN. Moreover, it needs to be investigated whether such attractor dynamics are present in the neural recordings from mice performing the change point detection task.

The RNNs in our study were trained via supervised learning. However, in real life, animals most likely learn a reversal learning task via reinforcement learning (RL), i.e., learn the task from reward outcomes. Neuromodulators play a key role in mediating RL in the brain. In a recent study, dopamine-based RL was used to train artificial RNNs to conduct reversal learning tasks. It was shown that neural activity in RNNs and mice performing the same tasks were in good agreement (*Wang et al., 2018*). In addition, projections of serotonin from dorsal raphe nuclei (*Hyun et al., 2023*; *Matias et al., 2017*) and norepinephrine from the locus coeruleus (*Su and Cohen, 2022*) to the cortical areas were shown to be involved in reversal learning. Further studies with biologically plausible network models, including neuromodulatory effects (*Harkin et al., 2023b*; *Wert-Carvajal et al., 2022*) or formal RL theories incorporating neuromodulators (*Harkin et al., 2023a*) could provide further insights into the role of neuromodulators in reversal learning.

## Conclusion

Our findings show that, when performing a reversal learning task, a cortical circuit represent reversal probability, not only in stable stationary states as in a line attractor model, but also in dynamic neural trajectories that can accommodate non-stationary task-related behaviors necessary for the task.

Such neural mechanism demonstrates the temporal flexibility of cortical computation and opens the opportunity for extending existing neural model for evidence accumulation by augmenting temporal dynamics.

## Methods

### Recurrent neural network

#### Network model

For the network model, we considered a continuous time recurrent neural network that operates in a dynamic regime relevant to cortical circuits. A strongly recurrent network with balanced excitation and inhibition has been known to capture canonical features of cortical neural activity, such as fluctuating activity and large trial-to-trial variability (**van Vreeswijk and Sompolinsky, 1996**; **Sompolinsky et al., 1988**; **Kadmon and Sompolinsky, 2015**). A standard network model for such balanced state contains both excitatory and inhibitory neurons, and its network dynamics have been investigated extensively (**van Vreeswijk and Sompolinsky, 1996**; **Renart et al., 2010**; **Brunel, 2000**).

In our study, we considered a network consisting of recurrently connected inhibitory neurons only. There were no excitatory neurons. Instead, constant excitatory external input was applied to inhibitory neurons to sustain the network activity. A purely inhibitory network can also operate in the balanced regime, where external excitatory input and recurrent inhibitory activity balance each other (**Kadmon and Sompolinsky, 2015**; **Brunel and Hansel, 2006**).

The main reason for adopting a purely inhibitory network was due to the GPU memory issue when training RNNs to perform the reversal learning task. As detailed in Methods Training scheme below, the RNNs are first simulated over all the trials in a block and then backpropagation-through-time is applied to update the connection weights. Since a block consists of many trials, the long duration of a block limits the size of network that can be trained. We observed that if connection probability is $p = 0.1$, and the population size of each neuron type is relatively large, then the excitatory-inhibitory network causes GPU memory overflow. This led us to consider a purely inhibitory network that operates in the balanced regime. Throughout network training, the signs of synaptic weights were preserved, resulting in a trained network that had only inhibitory synaptic connections.

The network dynamics were governed by the following equation

$$\tau \frac{d\mathbf{u}}{dt} = -\mathbf{u} + W^{rec}\phi(\mathbf{u}) + \mathbf{I}_{base} + \mathbf{I}_{cue} + \mathbf{I}_{feedback} \tag{1}$$

and the network readout was

$$z = \mathbf{w}^{out} \cdot \phi(\mathbf{u}). \tag{2}$$

Here, $\tau$ is the intrinsic time constant of the RNN. $\mathbf{u} \in \mathbb{R}^N$ is the neural activity of population of $N$ neurons. $W^{rec}$ is an $N \times N$ recurrent connectivity matrix with inhibitory synaptic weights: $W_{ij}^{rec} < 0$ for all connection from neuron $j$ to neuron $i$. The activation function is sigmoidal, $\phi(x) = 1/(1 + \exp[-(ax + b)])$, and is applied to $\mathbf{u}$ elementwise in $\phi(\mathbf{u})$. The baseline input $\mathbf{I}_{base}$ is constant in time and same for all neurons, the cue $\mathbf{I}_{cue}$ is turned on to signal the RNN to make a choice, and the feedback $\mathbf{I}_{feedback}$ provides information about the previous trial's choice and reward outcome (see **Table 1**).

The duration of a trial was $T = 500\,\text{ms}$. The intrinsic time constant $\tau = 20\,\text{ms}$ was significantly shorter than the trial duration. The feedback $\mathbf{I}_{feedback}$ was applied on the time interval $[0, T_{feedback}]$, and the cue $\mathbf{I}_{cue}$ was applied on the time interval $[T_{cue}^{start}, T_{cue}^{end}]$ where $T_{feedback} = 300\,\text{ms}$ and $T_{cue}^{start} = 250\,\text{ms}$, $T_{cue}^{end} = 300\,\text{ms}$. The feedback and cue overlapped during the cue period. The network choice was defined using the average of the readout $z$ on the time interval $[T_{choice}^{start}, T_{choice}^{end}]$ where $T_{choice}^{start} = 350\,\text{ms}$ and $T_{choice}^{end} = 400\,\text{ms}$ (see **Figure 1A**). The feedback $\mathbf{I}_{feedback}$ and cue $\mathbf{I}_{cue}$ were random vectors where each element was sample from Gaussian distribution with mean zero and standard deviation 0.9.

#### Reduced model

One-dimensional reduction of the network dynamics in a subspace defined by a task vector, $\mathbf{v}$, was derived as follows (see **Figure 5**). The projection of network activity onto the task vector was

**Table 1.** Four types of feedback inputs.

| Network choice | Rewarded choice | Feedback |
| --- | --- | --- |
| | A | A1 (rewarded) |
| A | B | A0 (no reward) |
| | A | B0 (no reward) |
| B | B | B1 (rewarded) |

$$x = \langle \phi(\mathbf{u}), \mathbf{v} \rangle. \tag{3}$$

Then, the dynamics of the projected activity is governed by

$$\tau \frac{dx}{dt} = \left\langle \nabla_{\mathbf{u}} \phi(\mathbf{u}) \cdot \frac{d\mathbf{u}}{dt}, \mathbf{v} \right\rangle = x_{rec} + x_{ext} \tag{4}$$

where

$$x_{rec} = \left\langle \nabla_{\mathbf{u}} \phi(\mathbf{u}) \cdot [-\mathbf{u} + W^{rec} \phi(\mathbf{u})], \mathbf{v} \right\rangle \tag{5}$$

$$x_{ext} = \left\langle \nabla_{\mathbf{u}} \phi(\mathbf{u}) \cdot \mathbf{I}, \mathbf{v} \right\rangle. \tag{6}$$

Here, $x_{rec}$ includes both the decay and recurrent terms, and $x_{ext}$ accounts for all external inputs $\mathbf{I} = \mathbf{I}_{base} + \mathbf{I}_{cue} + \mathbf{I}_{feedback}$.

## Reversal learning task

### Experiment setup for monkeys

The experimental setup for the animals was reported in a previous work (**Bartolo and Averbeck, 2020**). Here, we provide a summary of the behavioral task and neural recordings.

Two animals performed the reversal learning task in blocks of trials over eight sessions. In each trial, the animals were required to fixate centrally for a variable time (400 - 800 ms), and, subsequently, two cues were presented to the left and right of fixation dot. The animals made a saccade to select a target, hold sight for 500 ms, and reward for the choice was delivered stochastically. In What blocks, one image was designated as the high-value option, while the other image was designated as the low-value option at the beginning of a block, regardless of the location of the images. The high-value option was rewarded 70% of the time when chosen, and the low-value option was rewarded 30% of the time when chosen. In Where blocks, one location (e.g. left) was designated as the high-value option, while the other location (e.g. right) was designated as the low-value option, regardless of the actual images at those locations. Each block consisted of 80 trials. The reward probability of two options was switched at a random trial, within 20 trials centered around the mid-trial of a block. The animals explored available option to identify the block types and best options. The animals' choice (i.e. object location and identity) and the reward outcome (i.e. rewarded or not rewarded) in all the trials were recorded for further analysis.

The extracellular activity of neural populations was recorded in the dorso-lateral prefrontal cortex from both hemispheres, using eight multielectrode arrays while the monkeys performed the task. The size of neuronal populations had a range of 573 to 1023 with a median 706. The recorded neurons were evenly distributed across left and right hemispheres. To analyze the spiking activity $y_{it}(k)$ of neuron $i$ at time $t$ and trial $k$, a 300 ms-wide time window centered at time $t$ was slided forward in time with 20 ms increment, as the number of spikes neuron $i$ emitted in each time window was counted.

### Training setup for RNNs

*Overview*: To train the network, we used a block consisting of $T = 24$ trials. For testing the trained RNNs, as described in the main text, we expanded the number of trials in a block to 36 trials (see **Figure 1** for an example block). The reversal trial $r$ was sampled randomly and uniformly from 10 trials around the midtrial:.

$$r \in Unif[T_m - 5, T_m + 5] \quad \text{where} \quad T_m = T/2.$$

The network made a choice in each trial: A or B. To model which choice was rewarded, we generated a 'rewarded' choice for each trial. One of the choices was more likely to be rewarded than the other. The network's choice was compared to the rewarded choice, and the network received a feedback that signaled its choice and reward outcome (e.g., chose A and received a reward). The option that yielded higher reward prior to the reversal trial was switched to the other option at the reversal trial.

To train the network to reverse its preferred choice, we used the output of an ideal Bayesian observer model as teaching signal. Specifically, we first inferred the scheduled reversal trial (i.e. the trial at which reward probability switched) using the Bayesian model. Then, the network was trained to flip its preferred choice a few trials after the inferred scheduled reversal trial, such that network's behavioral reversal trial occurred a few trials after the scheduled reversal trial.

Note that, although we refer to 'rewarded' choices, there were no actual rewards in our network model. The 'rewarded' choices were set up to define feedback inputs that mimic the reward outcomes monkey received.

## Experiment variables

The important variables for training the RNN were network choice, rewarded choice and feedback.

### Network choice

To define network choice, we symmetrized the readout $\mathbf{z}^{sym} = (z, -z)$ and computed its log-softmax $f(\mathbf{z}^{sym}) = (\frac{e^z}{s}, \frac{e^{-z}}{s}) \equiv (f_0, f_1)$ where $s = e^z + e^{-z}$. The network choice was

$$c = \arg\max(f_0, f_1) = \begin{cases} A & \text{if} \quad f_0 > f_1 \\ B & \text{if} \quad f_0 < f_1 \end{cases} \tag{7}$$

where

$$A = 0, B = 1. \tag{8}$$

### Rewarded choice

To model stochastic rewards, rewarded choices were generated probabilistically for each trial $k$:

$$P_r(\text{rewarded choice} = A) = q_k$$
$$P_r(\text{rewarded choice} = B) = 1 - q_k. \tag{9}$$

The reversal of reward schedule was implemented by switching the target probability at the scheduled reversal trial of the block, denoted by $r_{sch}$.

$$(\text{before reversal}) \, q_k = p \text{ for } k < r_{sch}$$
$$(\text{after reversal}) \, q_k = 1 - p \text{ for } k \geq r_{sch}. \tag{10}$$

### Feedback

We considered that reward is delivered when the network choice agreed with the rewarded choice, and no reward is delivered when they disagreed. This led to four types of feedback inputs show in **Table 1**.

### Bayesian inference model

Here we formulate Bayesian models that infer the scheduled reversal trial or the behavior reversal trial.

#### Ideal observer model

The ideal observer model, developed previously (**Costa et al., 2015**; **Bartolo and Averbeck, 2020**), inferred the scheduled reversal trial and assumed that (a) the target probability was known (**Equation 9**) and (b) it switched at the reversal trial (**Equation 10**).

The data available to the ideal observer were the choice $y_k \in \{A, B\}$ and the reward outcome $z_k \in \{0, 1\}$ at all the trials $k \in [1, T]$. We inferred the posterior distribution of scheduled reversal at trials $k \in [1, T]$. By Bayes' rule

$$p(r|y, z) = p(y, z|r)p(r)/p(y, z). \tag{11}$$

We evaluated the posterior distribution of $r$ when data were available up to any trial $t \le T$. The likelihood function $f_{IO}(r) = p(y_{1:t}, z_{1:t}|r)$ of the ideal observer was defined by

$$f_{IO}(r) = \prod_{k=1}^{t} q_k.$$

For $k < r$,

$$q_k = p \quad \text{if} \quad y_k = A, \; z_k = 1 \tag{12}$$
$$= 1 - p \quad \text{if} \quad y_k = A, \; z_k = 0 \tag{13}$$
$$= 1 - p \quad \text{if} \quad y_k = B, \; z_k = 1 \tag{14}$$
$$= p \quad \text{if} \quad y_k = B, \; z_k = 0. \tag{15}$$

For $k \ge r$,

$$q_k = 1 - p \quad \text{if} \quad y_k = A, \; z_k = 1 \tag{16}$$
$$= p \quad \text{if} \quad y_k = A, \; z_k = 0 \tag{17}$$
$$= p \quad \text{if} \quad y_k = B, \; z_k = 1 \tag{18}$$
$$= 1 - p \quad \text{if} \quad y_k = B, \; z_k = 0. \tag{19}$$

To obtain the posterior distribution of $r$ (**Equation 11**), the likelihood function $f_{IO}(r)$ was evaluated for all $r \in [1, t]$, assuming flat prior $p(r)$ and normalizing by the choice and reward data $p(y_{1:t}, z_{1:t})$.

## Behavioral model

To infer the trial at which choice reversed, i.e., behavior reversal, we used a likelihood function that assumed the preferred choice probability switched at the behavior reversal. Here, the reward schedule was not known.

The data available to the behavioral model were the choice $y_k \in \{A, B\}$ at all the trials $k \in [1, T]$. We inferred the posterior distribution of behavior reversal at trials $k \in [1, T]$. By Bayes' rule

$$p(r|y) = p(y|r)p(r)/p(y). \tag{20}$$

The likelihood function for the behavioral model was

$$f_{BM}(r) = \prod_{k=1}^{t} q_k.$$

For $k < r$,

$$q_k = p \quad \text{if} \quad y_k = A$$
$$= 1 - p \quad \text{if} \quad y_k = B. \tag{22}$$

For $k \ge r$,

$$q_k = 1 - p \quad \text{if} \quad y_k = A$$
$$= p \quad \text{if} \quad y_k = B. \tag{24}$$

To obtain the posterior distribution of $r$, we assumed flat prior $p(r)$, as in the ideal observer, and normalized by the choice data $p(y_{1:t})$.

## Training scheme

### Overview

The ideal observer successfully inferred a scheduled reversal trial, which occurred randomly around the mid-trial. To learn to switch its preferred choice, we trained the network to learn from scheduled reversal trials inferred from the ideal observer. In other words, in a block consisting of $T$ trials, the

network choices and reward outcomes were fed into the ideal observer model to infer the randomly chosen scheduled reversal trial. Then, the network was trained to switch its preferred choice a few trials after the inferred reversal trial. This delay in the behavior reversal from the scheduled reversal was observed in monkey's reversal behavior (**Bartolo and Averbeck, 2020**) and a running estimate of the Maximum a Posterior of the reversal probability (see Step 3 below). As the inferred scheduled reversal trial varied across blocks, the network learned to reverse its choice in a block-dependent manner.

Below, we described the specific steps taken to train the network.

*Step 1.* Simulate the network starting from a random initial state, apply the external inputs, i.e., cue and feedback inputs, at each trial, and store the network choices and reward outcomes at all the trials in a block. The network dynamics is driven by the external inputs applied periodically over the trials.

*Step 2.* Apply the ideal observer model to network's choice and reward data to infer the scheduled reversal.

*Step 3.* Identify the trial $t^*$ at which network choice should be reversed.

The main observation is that the running estimate of Maximum a Posterior (MAP) of the reversal probability obtained from the ideal observer model converges a few trials past the MAP estimate. In other words, let

$$\text{MAP estimate}: \quad r_T^* = \underset{k \in [1,T]}{\arg\max}\, p(r = k | D_{1:T})$$

$$\text{Running estimate}: \quad r_t^* = \underset{k \in [1,t]}{\arg\max}\, p(r = k | D_{1:t})$$

then,

$$r_t^* \to r_T^* \quad \text{if} \quad t > r_T^*$$

where the convergence occurs around

$$t^* = r_T^* + \delta \quad \text{with} \quad \delta = 4.$$

This observation can be interpreted as follows. If a subject performing the reversal learning task employs the ideal observer model to detect the trial at which the reward schedule is reversed, the subject can infer the reversal of reward schedule a few trials past the actual reversal and then switch its preferred choice. This delay in behavioral reversal, relative to the reversal of reward schedule, is consistent with the monkeys switching their preferred choice a few trials after the reversal of reward schedule (**Bartolo and Averbeck, 2020**).

*Step 4.* Construct the choice sequences the network will learn.

We used the observation from Step 3 to define target choice outputs that switch abruptly a few trials after the reversal of reward schedule, denoted as $t^*$ in the following. In each block, the high-value option at the start of a trial was selected randomly between two options. If a block had $t^{(*)}$ as its initial high-value option, the target choice outputs were

$$\text{choice}_{IO}(k) = A \quad \text{if} \quad k < t^*$$

$$\text{choice}_{IO}(k) = B \quad \text{if} \quad k \geq t^*$$

On the other hand, if a block had $B$ as its initial high-value option, the target choice outputs were

$$\text{choice}_{IO}(k) = B \quad \text{if} \quad k < t^*$$

$$\text{choice}_{IO}(k) = A \quad \text{if} \quad k \geq t^*$$

An example of target choice outputs with $A$ as its initial high-value option is shown in **Figure 1B**.

*Step 5.* Define the loss function of a block.

$$loss = \sum_{k=1}^{T} CrossEnt\left(\text{choice}_{actual}(k), \text{choice}_{IO}(k)\right)$$

*Step 6.* Train the recurrent connectivity weights $W^{rec}$ and the readout weights $\mathbf{w}^{out}$ with back-propagation using Adam optimizer with learning rate $10^{-2}$. The learning rate was decayed by a factor 0.9 every 3 epochs. The batch size (i.e. the number of networks trained) was 256. The training was continued until the fraction of rewarded trials was close to reward probability $p$ of the preferred option.

## Targeted dimensionality reduction

Targeted dimensionality reduction (TDR) identifies population vectors that encode task variables explicitly or implicitly utilized in the experiment the subject or RNN performs (*Mante et al., 2013*). In this study, we were interested in identifying population vectors that encode choice preference and reversal probability. Once those task vectors were identified, we analyzed the neural activity projected to those vectors to investigate neural representation of task variables.

We describe how TDR was performed in our study (see *Mante et al., 2013* for the original reference). First, we regressed the neural activity of each neuron at each time point onto task variables of interest. Then we used the matrix of regression coefficients (i.e. neuron by time) to identify the task vector. Let $y_{it}(k)$ be the spiking rate of neuron $i$ at time $t$ on trial $k$ where we have $N$ neurons and $M$ time points. We regressed the spiking activity on task variables of interest $z^v(k)$ where the task variables were $v \in \{$reversal probability, choice preference, direction, object, block type, reward outcome, trial number$\}$. For each neuron-time pair, $(i, t)$, we performed linear regression over all trials $k \in [0, T]$ with a bias:

$$y_{it}(k) = \sum_v \beta_{it}^v z^v(k) + bias_{it}. \tag{25}$$

This regression analysis yielded an $N \times M$ coefficient matrix $\beta_{it}^v$ for each task variable, $v$. We considered this coefficient matrix as a population vector evolving in time: $\boldsymbol{\beta}_t^v = (\beta_{1t}^v, ..., \beta_{Nt}^v)$. Then, a task vector was defined as the population vector $\mathbf{w}^v \in \mathbb{R}^N$ at which the $L_2$-norm $\|\boldsymbol{\beta}_t^v\|_2$ achieved its maximum:

$$\mathbf{w}^v = \boldsymbol{\beta}_{t_{\max}}^v$$
$$t_{\max} = \arg \max_t \left\| \boldsymbol{\beta}_t^v \right\|_2. \tag{26}$$

We performed QR-decomposition on the matrix of task vectors $W = [\mathbf{w}_{rev}, \mathbf{w}_{choice}, ...]$ to orthogonalize the task vectors. Then, the population activity was projected onto each (orthogonalized) task vector to obtain the neural activity encoding each task variable:

$$x_t^v(k) = \mathbf{w}^v \cdot \mathbf{y}_t(k) \tag{27}$$

where $\mathbf{y}_t(k) = (y_{1t}(k), ..., y_{Nt}(k))$ is the population activity at time $t$ on trial $k$.

## Reward integration equation

To derive the reward integration equation shown in *Figure 3*, we considered the neural activity in a subspace encoding the reversal probability:

$$x_{rev}^k(t) = \mathbf{w}^{rev} \cdot \mathbf{y}^k(t). \tag{28}$$

We analyzed the neural activity at the time of cue onset $t = t_{on}$ and obtained a sequence of reversal probability activity across trials: $x_{rev}^0(t_{on}), \dots, x_{rev}^K(t_{on})$. To set up the reward integration equation

$$x_{pred}^{k+1}(t_{on}) = x_{pred}^k(t_{on}) + R_k^\pm(t_{on}) \quad \text{with} \quad x_{pred}^0(t_{on}) = x_{rev}^0(t_{on}), \tag{29}$$

we estimated the update $R_k^\pm(t_{on})$ driven by reward outcomes at each trial $k$. Specifically, the update term was defined as the block-average of the difference of reversal probability activity at adjacent trials:

$$\Delta x_{\text{rev}}^k(t_{\text{on}}) = x_{\text{rev}}^{k+1}(t_{\text{on}}) - x_{\text{rev}}^k(t_{\text{on}})$$

$$R_k^+(t_{\text{on}}) = \left\langle \Delta x_{\text{rev}}^k(t_{\text{on}}) \right\rangle_{b_k^+} \quad \text{if rewarded at trial } k \tag{30}$$

$$R_k^-(t_{\text{on}}) = \left\langle \Delta x_{\text{rev}}^k(t_{\text{on}}) \right\rangle_{b_k^-} \quad \text{if not rewarded at trial } k.$$

Here, $b_k^+$ denotes all the blocks across sessions (or networks) in which reward was received at trial $k$. Similarly, $b_k^-$ denotes all the blocks in which reward was not received at trial $k$.

To predict $x_{rev}^k(t_{on})$, we set the initial value $x_{pred}^0(t_{on}) = x_{rev}^0(t_{on})$ at trial 0 and sequentially predicted the following trials using *Equation 29* with the update term from *Equation 30*. The same analysis was performed at different time points $t$. We derived integration equations for each time and assessed its prediction accuracy as shown in *Figure 3F*.

To evaluate the contribution of reward and no-reward outcomes on the average responses of $\Delta x_{rev}^k(t)$ over blocks, we computed

$$\langle \Delta x_{rev}^k(t) \rangle_{b_k} = f_k^+ \langle \Delta x_{rev}^k(t) \rangle_{b_k^+} + f_k^- \langle \Delta x_{rev}^k(t) \rangle_{b_k^-} = f_k^+ R_k^+(t) + f_k^- R_k^-(t) \tag{31}$$

where

$$f_k^+ = \frac{|b_k^+|}{|b_k|}, \quad f_k^- = \frac{|b_k^-|}{|b_k|} \tag{32}$$

with $b_k = b_k^+ \cup b_k^-$ denote the fractions of reward and no-reward blocks at trial $k$. In *Figure 5D* and *Figure 6A*, the weighted responses, i.e., $f_k^+ R_k^+(t)$ and $f_k^- R_k^-(t)$, were shown.

## Contraction factor of reversal probability activity

We defined a contraction factor to quantify whether the reversal probability activity $x_{rev}(t)$ contracts to or diverges from its mean activity on a short time interval. The contraction factor on the $n^{th}$ time interval $[nL, (n+1)L]$ of length $L$ was defined to be the coefficient of a one-dimensional autoregressive equation the mean-centered reversal probability activity $z_{rev}^{(n)}(t)$ satisfied.

$$z_{rev}^{(n)}(t+1) = a z_{rev}^{(n)}(t) \quad \text{for} \quad t = 0, ..., L-1$$

where

$$z_{rev}^{(n)}(t) = x_{rev}(nL + t) - m$$

$$m = \frac{1}{L} \sum_{t=0}^{L} x_{rev}(nL + t).$$

The contraction factor $a$ of a time interval was estimated by performing a scalar linear regression without an intercept given the input data $(z_{rev}^{(n)}(0), \ldots, z_{rev}^{(n)}(L-1))$ and the output data $(z_{rev}^{(n)}(1), \ldots, z_{rev}^{(n)}(L))$.

## Generating the PFC reversal probability trajectories from initial states

We investigated if the PFC reversal probability trajectories $x_{rev}^k(t)$ of trial $k$ defined on the trial duration, $t \in [T_0, T_f]$, can be generated from their initial states (see *Figure 4D*). To test this idea, we trained support vector regression (SVR) model on the spiking activity of PFC neurons. The training data consisted of reversal probability trajectories $x_{rev}^k(t)$, $t \in [T_0, T_f]$ of 20 trials around the behavioral reversal trial in each block. About 50% of the blocks in an experiment session (specifically, 10 blocks) was randomly selected for training, and the remaining 50% of the blocks in the same session was used for testing. This procedure was repeated 10 times, training a different SVR model each time, to test models trained on different sets of blocks. For each experiment session, a different SVR model was trained.

To train an SVR model, reversal probability trajectory at each trial $k$ was divided into initial state and remaining trajectory:

$$\mathbf{x}_{init}^k = x_{rev}^k(T_0 : T_i) \in \mathbb{R}^{T_i - T_0 + 1} \tag{33}$$

$$\mathbf{x}_{traj}^k = x_{rev}^k(T_i + 1 : T_f) \in \mathbb{R}^{T_f - T_i} \tag{34}$$

where $T_0 = -500\,\text{ms}$ (start of fixation) and $T_i = -300\,\text{ms}$ (end of initial state). Then, an SVR model $f$ was constructed that takes the initial state $\mathbf{x}_{init}^k$ and a time point $s$ as its inputs and produces an approximation of $\mathbf{x}_{traj}^k(s)$:

$$\mathbf{x}_{traj}^k(s) \equiv x_{rev}^k(T_i + s) \approx f(\mathbf{x}_{init}^k, s) \quad \text{for} \quad s \in 1, 2, \ldots, T_f - T_i. \tag{35}$$

As described above, we combined 20 trials around the reversal trial ($k = -10, \ldots, 10$) from 10 randomly selected blocks in a session (about 50% of blocks in a session) to train the SVR model. The radial basis function was used as a kernel, and the optimal hyperparameters of the SVR model were selected through cross-validation. We used the Python's sklearn package to implement SVR.

## Null model
To demonstrate the significance of initial states in generating the reversal probability trajectories, we trained a null SVR model using randomly shuffled initial states. To generate the training and testing data for this null model, we shuffled the initial states $\mathbf{x}_{init}^k$ of 20 trials around the reversal trial in a block, while keeping the remaining trajectories in each trial unchanged. In other words, the null SVR model was trained to generate the reversal probability trajectory of a trial using the initial state from a randomly chosen trial in the same block as the input. As described above, 50% of the blocks in a session was randomly selected for training and the remaining blocks were used for testing. This training and testing procedure was repeated 10 times.

## Perturbation experiments
### Perturbing RNN
To perturb the activity of an RNN, a perturbation stimulus was applied to the network together with the cue. The perturbation duration was 50 ms, identical to the cue duration (*Figure 7B*). The perturbation stimulus was one of $v_+$, $v_-$, $v_{rnd}$ where $v_+$ is the reversal probability population vector derived from targeted dimensionality reduction (*Equation 26*), $v_-$ has the opposite sign of $v_+$, and $v_{rnd}$ is a vector with random elements sampled from a Gaussian distribution with mean 0 and standard deviation identical to that of $v_+$. The perturbation stimulus was added to the network dynamic equation (*Equation 1*) as one of the external inputs at either trial number -2, -1, or 0. The strength of perturbation was varied by modulating a multiplicative factor on $v_+, v_-, v_{rnd}$ from 0.5 to 4.0.

### Residual PFC activity
The residual PFC activity of reversal probability $x_{rev}^k(t)$ and choice $x_{choice}^k(t)$ was defined as the deviation of their individual trial activity from their block-averaged activity at the same trial: $Res_{rev}^k(t) = x_{rev}^k(t) - \langle x_{rev}^k(t) \rangle_{b_k}$ and $Res_{choice}^k(t) = x_{choice}^k(t) - \langle x_{choice}^k(t) \rangle_{b_k}$ where $b_k$ denotes trial $k$ in all the blocks across sessions. We analyzed the mean residual activity over the time interval [0 ms, 500 ms], i.e., $\overline{Res_{rev}^k} = \langle Res_{rev}^k(t) \rangle_{t \in [0,500]}$ and $\overline{Res_{choice}^k} = \langle Res_{choice}^k(t) \rangle_{t \in [0,500]}$. Then, at each trial $k$, we fitted a linear model to characterize the relationship between $\overline{Res_{rev}^k}$ and $\overline{Res_{choice}^k}$ by analyzing their values across blocks. *Figure 7F* shows the slopes of the linear models at each trial.

## Decoding monkey's behavioral reversal trial
The PFC activity $x_t^{rev}(k)$ encoding reversal probability was used to decode the behavioral reversal trial at which monkey reversed its preferred choice (see *Figure 5—figure supplement 1*). Our analysis is similar to the Linear Discriminant Analysis (LDA) performed in a previous study (*Bartolo and Averbeck, 2020*) at a fixed time point. Here, we applied LDA to time points across a trial.

For training, 90% of the blocks were randomly selected to train the decoder and remaining 10% of the blocks were used for testing. This was repeated 20 times. Input data to LDA was $x_t^{rev}(k)$ of $\Delta k$ trials around the reverse trial, i.e., $k \in [k_{rev} - \Delta k, \ldots, k_{rev} + \Delta k]$ with $\Delta k = 10$. At each trial $k$, we took the activity vector $\mathbf{x}_{t_0}^{rev}(k) = (x_{t_0 - \Delta t}^{rev}(k), \ldots, x_{t_0 + \Delta t}^{rev}(k))$ around time $t_0$ with $\Delta t = 160\,\text{ms}$. The target output

of LDA was a one-hot vector $\mathbf{y}^{target}$, whose element was 1 at the reversal trial $k_{rev}$ and 0 at other trials. The following input-output shows dataset of a block used for training:

$$\text{Input:} \quad \left[ \mathbf{x}_{t_0}^{rev}(k_{rev} - \Delta k), \ldots, \mathbf{x}_{t_0}^{rev}(k_{rev}), \ldots, \mathbf{x}_{t_0}^{rev}(k_{rev} + \Delta k) \right] \tag{36}$$

$$\text{Output:} \quad \mathbf{y}^{target} \tag{37}$$

Here, $\mathbf{x}_{t_0}^{rev}(k) \in \mathbb{R}^{T_{dec}}$ where $T_{dec} = 2\Delta t/\Delta h + 1$ denotes the number of time points around $t_0$ with time increment $\Delta h = 20\,\text{ms}$, and the one-hot vector $\mathbf{y}^{target} = [0, \ldots, 1, \ldots, 0] \in \mathbb{R}^{K_{dec}}$ where $K_{dec} = 2\Delta k + 1$ denotes the number of trials around the reversal trial. As mentioned above, this analysis was repeated for time point $t_0$ across a trial.

## Acknowledgements

This research was supported by the Intramural Research Program of the National Institutes of Health: the National Institute of Diabetes and Digestive and Kidney Diseases (NIDDK) and the National Institute of Mental Health (NIMH). This work utilized the computational resources of the NIH HPC Biowulf cluster (https://hpc.nih.gov).

## Additional information

### Funding

| Funder | Grant reference number | Author |
| --- | --- | --- |
| National Institute of Diabetes and Digestive and Kidney Diseases | | Christopher M Kim Carson C Chow |
| National Institute of Mental Health | | Bruno B Averbeck |

The funders had no role in study design, data collection and interpretation, or the decision to submit the work for publication.

### Author contributions

Christopher M Kim, Conceptualization, Software, Formal analysis, Validation, Investigation, Visualization, Methodology, Writing – original draft, Writing – review and editing; Carson C Chow, Conceptualization, Supervision, Funding acquisition, Investigation, Writing – review and editing; Bruno B Averbeck, Conceptualization, Data curation, Formal analysis, Supervision, Investigation, Methodology, Writing – original draft, Writing – review and editing

### Author ORCIDs

Christopher M Kim ⓘ https://orcid.org/0000-0002-1322-6207
Carson C Chow ⓘ https://orcid.org/0000-0003-1463-9553
Bruno B Averbeck ⓘ https://orcid.org/0000-0002-3976-8565

Reviewer #1 (Public review): https://doi.org/10.7554/eLife.103660.3.sa1
Reviewer #2 (Public review): https://doi.org/10.7554/eLife.103660.3.sa2
Reviewer #3 (Public review): https://doi.org/10.7554/eLife.103660.3.sa3
Author response https://doi.org/10.7554/eLife.103660.3.sa4

## Additional files

### Supplementary files
MDAR checklist

## Data availability

The Python code for training RNNs is available in the following Github repository: https://github.com/chrismkkim/LearnToReverse (copy archived at *Kim, 2025*).

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
