## [Editor Report · eLife Assessment]

The findings of this study are **valuable**, offering insights into the neural representation of reversal probability in decision-making tasks, with potential implications for understanding flexible behavior in changing environments. The study contains interesting comparisons between neural data and models, including evidence for partial consistency with line attractor models in this probabilistic reversal learning task. However, it remains **incomplete** due to issues related to how the RNN training and the analysis of its dynamics, which renders the evidence as not complete.

---

## [Referee Report · Reviewer #1 (Public review)]

The authors aimed to investigate how the probability of a reversal in a decision-making task is computed in cortical neurons. They analyzed neural activity in the prefrontal cortex of monkeys and units in recurrent neural networks (RNNs) trained on a similar task. Their goal was to understand how the dynamical systems that implement computation perform a probabilistic reversal learning task in RNNs and nonhuman primates.

Major strengths and weaknesses:

Strengths:

(1) Integrative Approach: The study exemplifies a modern approach by combining empirical data from monkey experiments with computational modeling using RNNs. This integration allows for a more comprehensive understanding of the dynamical systems that implement computation in both biological and artificial neural networks.

(2) The focus on using perturbations to identify causal relationships in dynamical systems is a good goal. This approach aims to go beyond correlational observations.

(3) The revised manuscript provides a more nuanced interpretation of the dynamics, reconciling the observations with aspects of line attractor models.

Weaknesses:

(1) The use of targeted dimensionality reduction (TDR) to identify the axis determining reversal probability may not necessarily isolate the dimension along which the RNN computes reversal probability. This should be computed from the RNN update itself rather than through a readout of network variance. Depending on how this is formulated, it could be something like the Jacobian of the state update with respect to inputs at input onset and with respect to the state during relaxation dynamics. This is worth thinking through further. It's important to try to take advantage of access afforded by using RNNs rather than solely relying on analyses available to us in neural data.

Appraisal of aims and conclusions:

The authors have substantially revised their interpretation of the results to reconcile their findings with line attractor models. They now acknowledge that their observation of reward integration explaining reversal probability activity (x_rev) is compatible with line attractor models, which addresses one of my main concerns.

Their expanded analysis now differentiates between two activity modes: (1) substantial non-stationary dynamics during a trial (incompatible with line attractors) and (2) stationary and stable dynamics at trial start (compatible with point attractors and line attractor models). This dual characterization provides a more complete picture of the dynamical system and highlights the composability of dynamical features.

Likely impact and utility:

This work makes a stronger contribution to our understanding of how probabilistic information is represented in neural circuits with intervening behaviors. The augmented model that combines elements of attractor dynamics with non-stationary trajectories offers a more comprehensive framework for understanding neural computations in decision-making tasks.

The data and methods could be useful to the community. While the authors have improved their analysis of network dynamics, additional reverse engineering that takes full advantage of access to the RNN's update equations could further strengthen the work.

---

## [Referee Report · Reviewer #2 (Public review)]

Summary:

In this work the authors trained RNN to perform a reversal task also performed by animals while PFC activity is recorded. The authors devised a new method to train RNN on this type of reversal task, which in principle ensures that the behavior of the RNN matches the behavior of the animal. They then performed some analysis of neural activity, both RNN and PFC recording, focusing on the neural representation of the reversal probability and its evolution across trials. Given the analysis presented, it has been difficult for me to asses at which point RNN can reasonably be compared to PFC recordings.

Strengths:

Focusing on a reversal task, the authors address a challenge in RNN training, as they do not use a standard supervised learning procedure where the desired output is available for each trial. They propose a new way of doing that.

They attempt to confront RNN and neural recordings in behaving animals.

Weaknesses:

It would be nice to better articulate the analysis results of the two training set-ups (with and without 0 response during fixation). The dynamical system analysis is confusing, the notions of stationary and non-stationary dynamics and its relationship with attractors are puzzling. Is there a line attractor in one case (with inputs orthogonal to the integration direction being called back to the attractor, and reward input aligned with the stable direction)? In the other case, do we have a cylindrical attracting manifold on which activity circles around and is pushed along the axis of the cylinder by reward inputs? Which case is closest to the PFC recordings?

---

## [Referee Report · Reviewer #3 (Public review)]

Summary:

Kim et al. present a study of the neural dynamics underlying reversal learning in monkey PFC and neural networks. Their main finding is that neural activity during fixation resembles a line attractor storing the current belief of the reversal state of the task. This is followed by richer dynamics unfolding throughout the remainder of the trial, which eventually converge to a new point on the line attractor by the start of the next trial. The idea of studying neural dynamics throughout the task (including intervening behaviour) is interesting, and the data provides some insights into the neural dynamics driving reversal learning. The modelling seems to support the analyses, but both the modelling and analyses also leave several open questions.

Strengths:

The paper addresses an interesting topic of the neural dynamics underlying reversal learning in PFC, using a combination of biological and simulated data. Reversal learning has been studied extensively in neuroscience, but this paper takes a step further by analysing neural dynamics throughout the trials instead of focusing on just the evidence integration epoch.

The authors show some close parallels between the experimental data and RNN simulations, both in terms of behaviour and neural dynamics. The analyses of how rewarded and unrewarded trials differentially affect dynamics throughout the trials in RNNs and PFC were particularly interesting. This work has the potential to provide new insights into the neural underpinnings of reversal learning.

Weaknesses:

Data analyses:

While the analyses seem mostly sound, one shortcoming is that they are all aligned to the inferred reversal trial rather than the true experimental reversal trial. For example, the analyses showing that 'x_rev' decays strongly after the reversal trial, irrespective of the reward outcome, seem like they are true essentially by design. The choice to align to the inferred reversal trial also makes this trial seem 'special' (e.g. in Fig 2 & Fig 6A), but it is unclear whether this is a real feature of the data or an artifact of effectively conditioning on a change in behaviour. It would be useful to investigate whether any of these analyses differ when aligned to the true reversal trial. It is also unsurprising that x_rev increases before the reversal and decreases after the reversal (it is hard to imagine a system where this is not the case), yet all of Fig 6 and several other analyses are devoted to this point.

Most of the analyses focus on the dynamics specifically in the x_rev subspace, but a major point of the paper is to say that biological (and artificial) networks may also have to do other things at different times in the trial. If that is the case, it would be interesting to also ask what happens in other subspaces of neural activity, which are not specifically related to evidence integration or choice - are there other subspaces that explain substantial variance? Do they relate to any meaningful features of the experiment?

This is especially important when considering analyses trying to establish the presence (or absence) of attractor dynamics in the circuit. In particular, activity in the x_rev subspace both affects and depends on other subspaces of neural activity, so it is not as meaningful to analyse the dynamics of this subspace in isolation. It would e.g. have been preferable to analyse the early-trial dynamics in the full state space and then possibly projecting onto x_rev, rather than first projecting activity onto x_rev and then fitting a linear autoregressive model.

Modelling:

There are a number of surprising and non-standard modelling choices made in this paper. For example, the choice to only use inhibitory neurons is non-conventional and it is not clear whether and how this impacts the results. The inputs are also provided without any learnable input weights, which makes it harder to interpret the input-driven dynamics during the different phases of a trial.

It is surprising that the RNN is "trained to flip its preferred choice a few trials after the inferred scheduled reversal trial", with the reversal trial inferred by an ideal Bayesian observer. A more natural approach would be to directly train the RNN to solve the task (by predicting the optimal choice) and then investigating the emergent behaviour & dynamics. If the authors prefer their imitation learning approach, it is also surprising that the network is trained to predict the reversal trial inferred using Bayesian smoothing instead of Bayesian filtering.

Finally, it was surprising that the network is trained and tested with different block lengths (24 & 36 trials, respectively), and it is not mentioned whether or how this affects behaviour.

---

## [Author Response]

The following is the authors’ response to the original reviews

Main revision made to the manuscript

The main revision made to the manuscript is to reconcile our findings with the line attractor model. The revision is based on Reviewer 1’s comment on reinterpreting our results as a superposition of an attractor model with fast timescale dynamics. We expanded our analysis regime to the start of a trial and characterized the overall within-trial dynamics to reinterpret our findings.

We first acknolwedge that our results are not in contradiction with evidence integration on a line attractor. As pointed out by the reviewers, our finding that the integration of reward outcome explains the reversal probability activity x_rev (Figure 3) is compatible with the line attractor model. However, the reward integration equation is an algebraic relation and does not characterize the dynamics of reversal probability activity. So a closer analysis on the neural dynamics is needed to assess the feasibility of line attractor.

In the revised manuscript, we show that x_rev exhibits two different activity modes (Figure 4). First, x_rev has substantial non-stationary dynamics during a trial, and this non-stationary activity is incompatible with the line attractor model, as claimed in the original manuscript. Second, we present new results showing that x_rev is stationary (i.e., constant in time) and stable (i.e., contracting) at the start of a trial. These two properties of x_rev support that it is a point attractor at the start of a trial and is compatible with the line attractor model.

We further analyze how the two activity modes are linked (Figure 4, Support vector regression). We show that the non-stationary activity is predictable from the stationary activity if the underlying dynamics can be inferred. In other words, the non-stationary activity during a trial is generated by an underlying dynamics with the initial condition provided by the stationary state at the start of trial.

These results suggest an extension of the line attractor model where an attractor state at the start of a trial provides an initial condition from which non-stationary activity is generated during a trial by an underlying dynamics associated with task-related behavior (Figure 4, Augmented model).

The separability of non-stationary trajectories (Figure 5 and 6) is a property of the non-stationary dynamics that allows separable points in the initial stationary state to remain separable during a trial, thus making it possible to represent distinct probabilistic values in non-stationary activity.

This revised interpretation of our results (1) retains our original claim that the non-stationary dynamics during a trial is incompatible with the line attractor model and (2) introduces attractor state at the start of a trial which is compatible with the line attractor model. Our anlaysis shows that the two activity modes are linked by an underlying dynamics, and the attractor state serves as initial state to launch the non-stationary activity.

**Responses to the Public Reviews:**

**Reviewer # 1:**

(1) To provide better explanation of the reversal learning task and network training method, we added detailed description of RNN and monkey task structure (Result Section 1), included a schematic of target outputs (Figure1B), explained the rationale behind using inhibitory network model (Method Section 1) and explained the supervised RNN training scheme (Result Section 1). This information can also be found in the Methods.

(2) Our understanding is that the augmented model discussed in the previous page is aligned with the model suggested by Reviewer 1: “a curved line attractor, with faster timescale dynamics superimposed on this structure”. It is likely that the “fast” non-stationary activity observed during the trial is driven by task-related behavior, thus is transient. For instance, we do not observe such non-stationary activity in the inter-trial-interval when the task-related behavior is absent. For this reason, the non-stationary trajectories were not considered to be part of the attractor. Instead, they are transient activity generated by the underlying neural dynamics associated with task-related behavior. We believe such characterization of faster timescale dynamics is consistent with Reviewer 1’s view and wanted to clarify that there are two different activity modes.

(3) We appreciate the reviewers (Reviewer 1 and Reviewer 2) comment that TDR may be limited in isolating the neural subspace of interest. Our study presents what could be learned from TDR but is by no means the only way to interpret the neural data. It would be of future work to apply other methods for isolating task-related neural activities.

We would appreciate it if the reviewers could share thoughts on what other alternative methods could better isolate the reversal probability activity.

**Reviewer # 2:**

(1) (i) We respectfully disagree with Reviewer 2’s comment that “no action is required to be performed by neurons in the RNN”. In our network setup, the output of RNN learns to choose a sign (+ or -), as Reviewer 2 pointed out, to make a choice. This is how the RNN takes an action. It is unclear to us what Reviewer 2 has intended by “action” and how reaching a target value (not just taking a sign) would make a significant difference in how the network performs the task.

(ii) From Reviewer 2’s comment that “no intervening behavior is thus performed by neurons”, we noticed that the term “intervening behavior” has caused confusion. It refers to task-related behavior, such as making choices or receiving reward, that the subject must perform across trials before reversing its preferred choice. These are the behaviors that intervene the reversal of preferred choice. To clarify its meaning, in the revised manuscript, we changed the term to “task-related behavior” and put them in context. For example, in the Introduction we state that “However, during a trial, task-related behavior, such as making decisions or receiving feedback, produced …”

(iii) As pointed out by Reviewer 2, the lack of fixation period in the RNN could make differences in the neural dynamics of RNN and PFC, especially at the start of a trial. We demonstrate this issue in Result Section 4 where we analyze the stationary activity at the start of a trial. We find that fixating the choice output to zero before making a choice promotes stationary activity and makes the RNN activity more similar to the PFC activity.

**Reviewer #3:**

(1) (i) In the previous study (Figure 1 in [Bartolo and Averbeck ‘20]), it was shown that neural activity can predict the behavioral reversal trial. This is the reason we examined the neural activity in the trials centered at the behavioral reversal trial. We explained in Result Section 2 that we followed this line of analysis in our study.

(ii) We would like to emphasize that the main point of Figures 4 and 5 is to show the separability of neural trajectories: the entire trajectory shifts without overlapping. It is not obvious that high-dimensional neural population activity from two trials should remain separated when their activities are compressed into a one-dimensional subspace. The onedimensional activities can easily collide since their activities are compressed into a lowdimensional space. We revised the manuscript to bring out these points. We added an opening paragraph that discusses separability of trajectories and revised the main text to bring out the findings on separability.

(iii) We agree with Reviewer 3 that it would be interesting to look at what happens in other subspace of neural activity that are not related to reversal probability and characterize how different neural subspace interact with each. However, the focus of this paper was the reversal probability activity, and we’d consider these questions out of the scope of current paper. We point out that, using the same dataset, neural activity related to other experimental variables were analyzed in other papers [Bartolo and Averbeck ’20; Tang, Bartolo and Averbeck ‘21]

(2) (i) In the revised manuscript, we added explanation on the rational behind choosing inhibitory network as a simplified model for the balanced state. In brief, strong inhibitory recurrent connections with strong excitatory external input operates in the balanced state, as in the standard excitatory-inhibitory network. We included references that studied this inhibitory network. We also explained the technical reason (GPU memory) for choosing the inhibitory model.

(ii) We thank the reviewer for pointing out that the original manuscript did not mention how the feedback and cue were initialized. They were random vectors sample from Gaussian distribution. We added this information in the revised manuscript. In our opinion, it is common to use random external inputs for training RNNs, as it is a priori unclear how to choose them. In fact, it is possible to analyze the effects of random feedback on one-dimensional x_rev dynamics by projecting the random feedback vector to the reversal probability vector. This is shown in Figure 4F.

(iii) We agree that it would be more natural to train the RNN to solve the task without using the Bayesian model. We point out this issue in the Discussion in the revised manuscript.

**Recommendations for the authors:**

**Reviewer #1:**
(1) My understanding of network training was that a Bayesian ideal observer signaled target output based on previous reward outcomes. However, the authors never mention that networks are trained by supervised learning in the main text until the last paragraph of the discussion. There is no mention that there was an offset in the target based on the behavior of the monkeys in the main text. These are really important things to consider in the context of the network solution after training. I couldn't actually find any figure that presents the target output for the network. Did I miss something key here?

In Result Section 1, we added a paragraph that describes in detail how the RNN is trained. We explained that the network is first simulated and then the choice outputs and reward outcomes are fed into the Bayesian model to infer the scheduled reversal trial. A few trials are added to the inferred reversal trial to obtain the behavioral reversal trial, as found in a previous study [Bartolo and Averbeck ‘20]. Then the network weights are updated by backpropagation-through-time via supervised learning.

In the original manuscript, the target output for the network was described in Methods Section 2.5, Step 4. To make this information readily accessible, we added a schematic in Figure 1B that shows the scheduled, inferred and behavioral reversal trials. It also shows how the target choice ouputs are defined. They switch abruptly at the behavioral reversal trial.

(2) The role of block structure in the task is an important consideration. What are the statistics of block switches? The authors say on average the reversals are every 36 trials, but also say there are random block switches. The reviewer's notes suggest that both the networks and monkeys may be learning about the typical duration of blocks, which could influence their expectations of reversals. This aspect of the task design should be explained more thoroughly and considered in the context of Figure 1E and 5 results.

We provided more detailed description of the reversal learning task in Result Section 1. We clarified that (1) a task is completed by executing a block of fixed number of trials and (2) reversal of reward schedule occurrs at a random trial around the mid-trial in a block. The differences in the number of trials in a block that the RNNs (36) and the monkeys (80) perform are also explained. We also pointed out the differences in how the reversal trial is randomly sampled.

However, it is unclear what Reviewer 1 meant by random block switches. Our reversal learning task is completed when a block of fixed number of trials is executed. Reversal of reward schedule occurs only once on a randomly selected trial in the block, and the reversed reward schedule is maintained until the end of a block. It is different from other versions of reveral learning where the reward schedule switches multiple times across trials. We clarified this point in Result Section 1.

(3) The relationship between the supervised learning approach used in the RNNs and reinforcement learning was confused in the discussion. "Although RNNs in our study were trained via supervised learning, animals learn a reversal-learning task from reward feedback, making it into a reinforcement learning (RL) problem." This is fundamentally not true. In the case of this work, the outcome of the previous trial updates the target output, rather than the trial and error type learning as is typical in reinforcement learning. Networks are not learning by reinforcement learning and this statement is confusing.

We agree with Reviewer 1’s comment that the statement in the original manuscript is confusing. Our intention was to point out that our study used supervised learning, and this is different from animals learn by reinforcement learning in rea life. We revised the sentence in Discussion as follows:

“The RNNs in our study were trained via supervised learning. However, in real life, animals learn a reversal learning task via reinforcement learning (RL), i.e., learn the task from reward outcomes.”

(4) The distinction between line attractors and the dynamic trajectories described by the authors deserves further investigation. A significant concern arises from the authors' use of targeted dimensionality reduction (TDR), a form of regression, to identify the axis determining reversal probability. While this approach can reveal interesting patterns in the data, it may not necessarily isolate the dimension along which the RNN computes reversal probability. This limitation could lead to misinterpretation of the underlying neural dynamics.a) This manuscript cites work described in "Prefrontal cortex as a meta-reinforcement learning system," which examined a similar task. In that study, the authors identified a v-shaped curve in the principal component space of network states, representing the probability of choosing left or right.Importantly, this curve is topologically equivalent to a line and likely represents a line attractor. However, regressing against reversal probability in such a case would show that a single principal component (PC2) directly correlates with reversal probability.b) The dynamics observed in the current study bear a striking resemblance to this structure, with the addition of intervening loops in the network state corresponding to within-trial state evolution. Crucially, these observations do not preclude the existence of a line attractor. Instead, they may reflect the network's need to produce fast timescale dynamics within each trial, superimposed on the slower dynamics of the line attractor.c) This alternative interpretation suggests that reward signals could function as inputs that shift the network state along the line attractor, with information being maintained across trials. The fast "intervening behaviors" observed by the authors could represent faster timescale dynamics occurring on top of the underlying line attractor dynamics, without erasing the accumulated evidence for reversals.d) Given these considerations, the authors' conclusion that their results are better described by separable dynamic trajectories rather than fixed points on a line attractor may be premature. The observed dynamics could potentially be reconciled with a more nuanced understanding of line attractor models, where the attractor itself may be curved and coexist with faster timescale dynamics.

We appreciate the insightful comments on (1) the similarity of the work by Wang et al ’18 with our findings and (2) an alternative interpretation that augments the line attractor with fast timescale dynamics.

(1) We added a discussion of the work by Wang et al ’18 in Result Section 2 to point out the similarity of their findings in the principal component space with ours in the x_rev and x_choice space. We commented that such network dynamics could emerge when learning to perform the reversal learning the task, regardless of the training schemes.

We also mention that the RL approach in Wang et al ’18 does not consider within-trial dynamics, therefore lacks the non-stationary activity observed during the trial in the PFC of monkeys and our trained RNNs.

(2) We revised our original manuscript substantially to reconcile the line attractor model with the nonstationary activity observed during a trial.

Here are the highlights of the revised interpretation of the PFC and the RNN network activity

- The dynamics of x_rev consists of two activity modes, i.e., stationary activity at the start of a trial and non-stationary activity during the trial. Schematic of the augmented model that reconciles two activity modes is shown in Figure 4A. Analysis of the time derivative (dx_reverse / dt) and contractivity of the stationary state are shown in Figure 4B,C to demonstrate two activity modes.

- We discuss in Result Section 4 main text that the stationary activity is consistent with the line attractor model, but the non-stationary activity deviates from the model.

- The two activity modes are linked dynamically. There is an underlying dynamics that can map the stationary state to the non-stationary trajectory. This is shown by predicting the nonstationary trajectory with the stationary state using a support vector regression model. The prediction results are shown in Figure 4D,E,F.

- We discuss in Result Section 4 an extension of the standard line attractor model: points on the line attractor can serve as initial states that launch non-stationary activity associated with taskrelated behavior.

- The separability of neural trajectories presented in Result Section 5 is framed as a property of the non-stationary dynamics associated with task-related behavior.

To strengthen their claims, the authors should:(1) Provide a more detailed description of their RNN training paradigm and task structure, including clear illustrations of target outputs.(2) Discuss how their findings relate to and potentially extend previous work on similar tasks, particularly addressing the similarities and differences with the v-shaped state organization observed in reinforcement learning contexts. (https://www.nature.com/articles/s41593-018-0147-8 Figure1).(3) Explore whether their results could be consistent with a curved line attractor model, rather than treating line attractors and dynamic trajectories as mutually exclusive alternatives.

Our response to these three comments is described above.

Addressing these points would significantly enhance the impact of the study and provide a more nuanced understanding of how reversal probabilities are represented in neural circuits.In conclusion, while this study provides interesting insights into the neural representation of reversal probability, there are several areas where the methodology and interpretations could be refined.Additional Minor Concerns:(1) Network Training and Reversal Timing: The authors mention that the network was trained to switch after a reversal to match animal behavior, stating "Maximum a Posterior (MAP) of the reversal probability converges a few trials past the MAP estimate." More explanation of how this training strategy relates to actual animal behavior would enhance the reader's understanding of the meaning of the model's similarity to animal behavior in Figure 1.

In Method Section 2.5, we described how our observation that the running estimate of MAP converges a few trials after the actual MAP is analogous to the animal’s reversal behavior.

“This observation can be interpreted as follows. If a subject performing the reversal learning task employs the ideal observer model to detect the trial at which reward schedule is reversed, the subject can infer the reversal of reward schedule a few trials past the actual reversal and then switch its preferred choice. This delay in behavioral reversal, relative to the reversal of reward schedule, is analogous to the monkeys switching their preferred choice a few trials after the reversal of reward schedule.”

In Step 4, we also mentioned that the target choice outputs are defined based on our observation in Step 3.

“We used the observation from Step 3 to define target choice outputs that switch abruptly a few trials after the reversal of reward schedule, denoted as \begin{document}$t^{(*)}$\end{document} in the following. An example of target outputs are shown in Fig.\,\ref{fig_behavior}B.”

(2) How is the network simulated in step 1 of training? Is it just randomly initialized? What defines this network structure?

The initial state at the start of a block was random. We think the initial state is less relevant as the external inputs (i.e., cue and feedback) are strong and drive the network dynamics. We mentioned these setup and observation in Step 1 of training.

“Step 1. Simulate the network starting from a random initial state, apply the external inputs, i.e., cue and feedback inputs, at each trial and store the network choices and reward outcomes at all the trials in a block. The network dynamics is driven by the external inputs applied periodically over the trials.”

(3) Clarification on Learning Approach: More description of the approach in the main text would be beneficial. The statement "Here, we trained RNNs that learned from a Bayesian inference model to mimic the behavioral strategies of monkeys performing the reversal learning task [2, 4]" is somewhat confusing, as the model isn't directly fit to monkey data. A more detailed explanation of how the Bayesian inference model relates to monkey behavior and how it's used in RNN training would improve clarity.

We described the learning approach in more detail, but also tried to be concise without going into technical details.

We revised the sentence in Introduction as follows:

“We sought to train RNNs to mimic the behavioral strategies of monkeys performing the reversal learning task. Previous studies (Costa et al., 2015; Bartolo and Averbeck, 2020) have shown that a Bayesian inference model can capture a key aspect of the monkey's behavioral strategy, i.e., adhere to the preferred choice until the reversal of reward is detected and then switch abruptly. We trained the RNNs to replicate this behavioral strategy by training them on target behaviors generated from the Bayesian model.”

We also added a paragraph in Result Section 1 that explains in detail how the training approach works.

(4) In Figure 1B, it would be helpful to show the target output.

We added a figure in Fig1B that shows a schematic of how the target output is generated.

(5) An important point to consider is that a line attractor can be curved while still being topologically equivalent to a line. This nuance makes Figure 4A somewhat difficult to interpret. It might be helpful to discuss how the observed dynamics relate to potentially curved line attractors, which could provide a more nuanced understanding of the neural representations.

As discussed above, we interpret the “curved” activity during the trial as non-stationary activity. We do not think this non-stationary activity would be characterized as attractor. Attractor is (1) a minimal set of states that is (2) invariant under the dynamics and (3) attracting when perturbed into its neighborhood [Strogatz, *Nonlinear dynamics and chaos*]. If we consider the autonomous system without the behavior-related external input as the base system, then the non-stationary states could satisfy (2) and (3) but not (1), so they are not part of the attractor. If we include the behavior-related external input to the autonomous dynamics, then it may be possible that the non-stationary trajectories are part of the attractor. We adopted the former interpretation as the behavior-related inputs are external and transient.

(6) The results of the perturbation experiments seem to follow necessarily from the way x_rev was defined. It would be valuable to clarify if there's more to these results than what appears to be a direct consequence of the definition, or if there are subtleties in the experimental design or analysis that aren't immediately apparent.

The neural activity x_rev is correlated to the reversal probability, but it is unclear if the activity in this neural subspace is causally linked to behavioral variables, such as choice output. We added this explanation at the beginning of Results Section 7 to clarify the reason for performing the perturbation experiments.

“The neural activity $x_{rev}$ is obtained by identifying a neural subspace correlated to reversal probability. However, it remains to be shown if activity within this neural subspace is causally linked to behavioral variables, such as choice output.”

**Reviewer #2:**
Below is a list of things I have found difficult to understand, and been puzzled/concerned about while reading the manuscript:(1) It would be nice to say a bit more about the dataset that has been used for PFC analysis, e.g. number of neurons used and in what conditions is Figure 2A obtained (one has to go to supplementary to get the reference).

We added information about the PFC dataset in the opening paragraph of Result Section 2 to provide an overview of what type of neural data we’ve analyzed. It includes information about the number of recorded neurons, recording method and spike binning process.

(2) It would be nice to give more detail about the monkey task and better explain its trial structure.

In Result Section 1 we added a description of the overall task structure (and its difference with other versions of revesal learning task), the RNN / monkey trial structure and differences in RNN and monkey tasks.

(3) In the introduction it is mentioned that during the hold period, the probability of reversal is represented. Where does this statement come from?

The fact that neural activity during a hold period, i.e., fixation period before presenting the target images, encodes the probability of reversal was demonstrated in a previous study (Bartolo and Averbeck ’20).

We realize that our intention was to state that, during the hold period, the reversal probability activity is stationary as in the line attractor model, instead of focusing on that the probability of reversal is represented during this period. We revised the sentence to convey this message. In addition, we revised the entire paragraph to reinterpret our findings: there are two activity modes where the stationary activity is consistent with the line attractor model but the non-stationary activity deviates from it.

(4) "Around the behavioral reversal trial, reversal probabilities were represented by a family of rankordered trajectories that shifted monotonically". This sentence is confusing and hard to understand.

Thank you for point this out. We rewrote the paragraph to reflect our revised interpretation. This sentence was removed, as it can be considered as part of the result on separable trajectories.

(5) For clarity, in the first section, when it is written that "The reversal behavior of trained RNNs was similar to the monkey's behavior on the same task" it would be nice to be more precise, that this is to be expected given the strategy used to train the network.

We removed this sentence as it makes a blanket statement. Instead, we compared the behavioral outputs of the RNNs and the monkeys one by one.

We added a sentence in Result Section 1 that the RNN’s abrupt behavioral reversal is expected as they are trained to mimic the target choice outputs of the Bayesian model.

“Such abrupt reversal behavior was expected as the RNNs were trained to mimic the target outputs of the Bayesian inference model.”

(6) What is the value of tau used in eq (1), and how does it compare to trial duration?

We described the value of time constant tau in Eq (1) and also discussed in Result Section 1 that tau=20ms is much faster than trial duration 500ms, thus the persistent behavior seen in trained RNNs is due to learning.

(7) It would be nice to expand around the notion of « temporally flexible representation » to help readers grasp what this means.

Instead of stating that the separable dynamic trajectories have “temporally flexible representation”, we break down in what sense it is temporally flexible: separable dynamic trajectories can accommodate the effects that task-related behavior have on generating non-stationary neural dynamics.

“In sum, our results show that, in a probabilistic reversal learning task, recurrent neural networks encode reversal probability by adopting, not only stationary states as in a line attractor, but also separable dynamic trajectories that can represent distinct probabilistic values while accommodating non-stationary dynamics associated with task-related behavior.”

**Reviewer #3:**
(1) Data:It would be useful to describe the experimental task, recording setup, and analyses in much more detail - both in the text and in the methods. What part of PFC are the recordings from? How many neurons were recorded over how many sessions? Which other papers have they been used in? All of these things are important for the reader to know, but are not listed anywhere. There are also some inconsistencies, with the main text e.g. listing the 'typical block length' as 36 trials, and the methods listing the block length as 24 trials (if this is a difference between the biological data and RNN, that should be more explicit and motivated).

We provided more detailed description of the monkey experimental task and PFC recordings in Result Section 1. We also added a new section in Methods 2.1 to describe the monkey experiment.

The experimental analyses should be explained in more detail in the methods. There is e.g. no detailed description of the analysis in Figure 6F.

We added a new section in Methods 6 to describe how the residual PFC activity is computed. It also describes the RNN perturbation experiments.

Finally, it would be useful for more analyses of monkey behaviour and performance, either in the main text or supplementary figures.

We did not pursue this comment as it is unclear how additional behavioral analyses would improve the manuscript.

(2) Model:When fitting the network, 'step 1' of training in 2.3 seems superfluous. The posterior update from getting a reward at A is the same as that from not getting a reward at B (and vice versa), and it is therefore completely independent of the network choice. The reversal trial can therefore be inferred without ever simulating the network, simply by generating a sample of which trials have the 'good' option being rewarded and which trials have the 'bad' option being rewarded.

We respectfully disagree with Reviewer 3’s comment that the reversal trial can be inferred without ever simulating the network. The only way for the network to know about the underlying reward schedule is to perform the task by itself. By simulating the network, it can sample the options and the reward outcomes.

Our understanding is that Review 3 described a strategy that a human would use to perform this task. Our goal was to train the RNN to perform the task.

Do the blocks always start with choice A being optimal? Is everything similar if the network is trained with a variable initial rewarded option? E.g. in Fig 6, would you see the appropriate swap in the effect of the perturbation on choice probability if choice B was initially optimal?

Thank you for pointing out that the initial high-value option can be random. When setting up the reward schedule, the initial high-value option was chosen randomly from two choice outputs and, at the scheduled reversal, it was switched to the other option. We did not describe this in the original manuscript.

We added a descrption in Training Scheme Step 4 that the the initial high-value option is selected randomly. This is also explained in Result Section 1 when we give an overview of the RNN training procedure.

(3) Content:It is rarely explained what the error bars represent (e.g. Figures 3B, 4C, ...) - this should be clear in all figures.

We added that the error bars represent the standard error of mean.

Figure 2A: this colour scheme is not great. There are abrupt colour changes both before and after the 'reversal' trial, and both of the extremes are hard to see.

We changed the color scheme to contrast pre- and post-reversal trials without the abrupt color change.

Figure 3E/F: how is prediction accuracy defined?

We added that the prediction accuracy is based on Pearson correlation.

Figure 4B: why focus on the derivative of the dynamics? The subsequent plots looking at the actual trajectories are much easier to understand. Also - what is 'relative trial' relative to?

The derivative was analyzed to demonstrate stationarity or non-stationarity of the neural activity. We think it will be clearer in the revised manuscript that the derivative allows us to characterize those two activity modes.

Relative trial number indicate the trial position relative to the behavioral reversal trial. We added this description to the figures when “relative trial” is used.

Figure 4C: what do these analyses look like if you match the trial numbers for the shift in trajectories? As it is now, there will presumably be more rewarded trials early and late in each block, and more unrewarded trials around the reversal point. Does this introduce biases in the analysis? A related question is (i) why the black lines are different in the top and bottom plots, and (ii) why the ends of the black lines are discontinuous with the beginnings of the red/blue lines.

We could not understand what Reviewer 3 was asking in this comment. It’d help if Review 3 could clarify the following question:

“Figure 4C: what do these analyses look like if you match the trial numbers for the shift in trajectories?”

Question (i): We wanted to look at how the trajectory shifts in the subsequent trial if a reward is or is not received in the current trial. The top panel analyzed all the trials in which the subsquent trial did not receive a reward. The bottom panel analyzed all the trials in which the subsequent trial received a reward. So, the trials analyzed in the top and bottom panels are different, and the black lines (x_rev of “current” trial) in the top and bottom panels are different.

Question (ii): Black line is from the preceding trial of the red/blue lines, so if trials are designed to be continuous with the inter-trial-interval, then black and red/blue should be continuous. However, in the monkey experiment, the inter-trial-intervals were variable, so the end of current trial does not match with the start of next trial. The neural trajectories presented in the manuscript did not include the activity in this inter-trial-interval.

Figure 6C: are the individual dots different RNNs? Claiming that there is a decrease in Delta x_choice for a v_+ stimulation is very misleading.

Yes individual dots are different RNN perturbations. We added explanation about the dots in Figure7C caption.

We agree with the comment that \Delta x_choice did not decrease. This sentence was removed. Instead, we revised the manuscript to state that x_choice for v_+ stimulation was smaller than the x_choice for v_- stimulation. We performed KS-test to confirm statistical significance.

Discussion: "...exhibited behaviour consistent with an ideal Bayesian observer, as found in our study". The RNN was explicitly trained to reproduce an ideal Bayesian observer, so this can only really be considered an assumption (not a result) in the present study.

We agree that the statement in the original manuscript is inaccurate. It was revised to reflect that, in the other study, behavior outputs similar to a Bayesian observer emerged by simply learning to do the task, intead of directly mimicking the outputs of Bayesian observer as done in our study.

“Authors showed that trained RNNs exhibited behavior outputs consistent with an ideal Bayesian observer without explicitly learning from the Bayesian observer. This finding shows that the behavioral strategies of monkeys could emerge by simply learning to do the task, instead of directly mimicking the outputs of Bayesian observer as done in our study.”

Methods: Would the results differ if your Bayesian observer model used the true prior (i.e. the reversal happens in the middle 10 trials) rather than a uniform prior? Given the extensive literature on prior effects on animal behaviour, it is reasonable to expect that monkeys incorporate some non-uniform prior over the reversal point.

Thank you for pointing out the non-uniform prior. We haven’t conducted this analysis, but would guess that the convergence to the posterior distribution would be faster. We’d have to perform further analysis, which is out of the scope of this paper, to investigate whether the posteior distribution would be different from what we obtained from uniform prior.

Making the code available would make the work more transparent and useful to the community.

The code is available in the following Github repository: https://github.com/chrismkkim/LearnToReverse